# Newton-Stein Method:
# A Second Order Method for GLMs via Stein's Lemma

**Murat A. Erdogdu**
Department of Statistics
Stanford University
erdogdu@stanford.edu

## Abstract

We consider the problem of efficiently computing the maximum likelihood estimator in *Generalized Linear Models* (GLMs) when the number of observations is much larger than the number of coefficients ($n \gg p \gg 1$). In this regime, optimization algorithms can immensely benefit from approximate second order information. We propose an alternative way of constructing the curvature information by formulating it as an estimation problem and applying a *Stein-type lemma*, which allows further improvements through sub-sampling and eigenvalue thresholding. Our algorithm enjoys fast convergence rates, resembling that of second order methods, with modest per-iteration cost. We provide its convergence analysis for the case where the rows of the design matrix are i.i.d. samples with bounded support. We show that the convergence has two phases, a quadratic phase followed by a linear phase. Finally, we empirically demonstrate that our algorithm achieves the highest performance compared to various algorithms on several datasets.

## 1 Introduction

Generalized Linear Models (GLMs) play a crucial role in numerous statistical and machine learning problems. GLMs formulate the natural parameter in exponential families as a linear model and provide a miscellaneous framework for statistical methodology and supervised learning tasks. Celebrated examples include linear, logistic, multinomial regressions and applications to graphical models [MN89, KF09].

In this paper, we focus on how to solve the maximum likelihood problem efficiently in the GLM setting when the number of observations $n$ is much larger than the dimension of the coefficient vector $p$, i.e., $n \gg p$. GLM optimization task is typically expressed as a minimization problem where the objective function is the negative log-likelihood that is denoted by $\ell(\beta)$ where $\beta \in \mathbb{R}^p$ is the coefficient vector. Many optimization algorithms are available for such minimization problems [Bis95, BV04, Nes04]. However, only a few uses the special structure of GLMs. In this paper, we consider updates that are specifically designed for GLMs, which are of the from

$$\beta \leftarrow \beta - \gamma \mathbf{Q} \nabla_\beta \ell(\beta), \tag{1.1}$$

where $\gamma$ is the step size and $\mathbf{Q}$ is a scaling matrix which provides curvature information.

For the updates of the form Eq. (1.1), the performance of the algorithm is mainly determined by the scaling matrix $\mathbf{Q}$. Classical *Newton's Method* (NM) and *Natural Gradient Descent* (NG) are recovered by simply taking $\mathbf{Q}$ to be the inverse Hessian and the inverse Fisher's information at the current iterate, respectively [Ama98, Nes04]. Second order methods may achieve quadratic convergence rate, yet they suffer from excessive cost of computing the scaling matrix at every iteration. On the other hand, if we take $\mathbf{Q}$ to be the identity matrix, we recover the simple *Gradient Descent* (GD) method which has a linear convergence rate. Although GD's convergence rate is slow compared to that of second order methods, modest per-iteration cost makes it practical for large-scale problems.

The trade-off between the convergence rate and per-iteration cost has been extensively studied [BV04, Nes04]. In $n \gg p$ regime, the main objective is to construct a scaling matrix $\mathbf{Q}$ that

is computational feasible and provides sufficient curvature information. For this purpose, several Quasi-Newton methods have been proposed [Bis95, Nes04]. Updates given by Quasi-Newton methods satisfy an equation which is often referred as the *Quasi-Newton relation*. A well-known member of this class of algorithms is the *Broyden-Fletcher-Goldfarb-Shanno* (BFGS) algorithm [Nes04].

In this paper, we propose an algorithm that utilizes the structure of GLMs by relying on a Stein-type lemma [Ste81]. It attains fast convergence rate with low per-iteration cost. We call our algorithm *Newton-Stein Method* which we abbreviate as *NewSt*. Our contributions are summarized as follows:

- We recast the problem of constructing a scaling matrix as an estimation problem and apply a Stein-type lemma along with sub-sampling to form a computationally feasible $\mathbf{Q}$.
- Newton method's $\mathcal{O}(np^2 + p^3)$ per-iteration cost is replaced by $\mathcal{O}(np + p^2)$ per-iteration cost and a one-time $\mathcal{O}(n|S|^2)$ cost, where $|S|$ is the sub-sample size.
- Assuming that the rows of the design matrix are i.i.d. and have bounded support, and denoting the iterates of Newton-Stein method by $\{\hat{\beta}^t\}_{t \geq 0}$, we prove a bound of the form

$$\left\|\hat{\beta}^{t+1} - \beta_*\right\|_2 \leq \tau_1 \left\|\hat{\beta}^t - \beta_*\right\|_2 + \tau_2 \left\|\hat{\beta}^t - \beta_*\right\|_2^2, \tag{1.2}$$

  where $\beta_*$ is the minimizer and $\tau_1, \tau_2$ are the convergence coefficients. The above bound implies that the convergence starts with a quadratic phase and transitions into linear later.
- We demonstrate its performance on four datasets by comparing it to several algorithms.

The rest of the paper is organized as follows: Section 1.1 surveys the related work and Section 1.2 introduces the notations used throughout the paper. Section 2 briefly discusses the GLM framework and its relevant properties. In Section 3, we introduce Newton-Stein method, develop its intuition, and discuss the computational aspects. Section 4 covers the theoretical results and in Section 4.3 we discuss how to choose the algorithm parameters. Finally, in Section 5, we provide the empirical results where we compare the proposed algorithm with several other methods on four datasets.

## 1.1 Related work

There are numerous optimization techniques that can be used to find the maximum likelihood estimator in GLMs. For moderate values of $n$ and $p$, classical second order methods such as NM, NG are commonly used. In large-scale problems, data dimensionality is the main factor while choosing the right optimization method. Large-scale optimization tasks have been extensively studied through online and batch methods. Online methods use a gradient (or sub-gradient) of a single, randomly selected observation to update the current iterate [Bot10]. Their per-iteration cost is independent of $n$, but the convergence rate might be extremely slow. There are several extensions of the classical stochastic descent algorithms (SGD), providing significant improvement and/or stability [Bot10, DHS11, SRB13].

On the other hand, batch algorithms enjoy faster convergence rates, though their per-iteration cost may be prohibitive. In particular, second order methods attain quadratic rate, but constructing the Hessian matrix requires excessive computation. Many algorithms aim at forming an approximate, cost-efficient scaling matrix,. This idea lies at the core of Quasi-Newton methods [Bis95].

Another approach to construct an approximate Hessian makes use of sub-sampling techniques [Mar10, BCNN11, VP12, EM15]. Many contemporary learning methods rely on sub-sampling as it is simple and it provides significant boost over the first order methods. Further improvements through conjugate gradient methods and Krylov sub-spaces are available.

Many hybrid variants of the aforementioned methods are proposed. Examples include the combinations of sub-sampling and Quasi-Newton methods [BHNS14], SGD and GD [FS12], NG and NM [LRF10], NG and low-rank approximation [LRMB08]. Lastly, algorithms that specialize on certain types of GLMs include coordinate descent methods for the penalized GLMs [FHT10] and trust region Newton methods [LWK08].

## 1.2 Notation

Let $[n] = \{1, 2, ..., n\}$, and denote the size of a set $S$ by $|S|$. The gradient and the Hessian of $f$ with respect to $\beta$ are denoted by $\nabla_\beta f$ and $\nabla_\beta^2 f$, respectively. The $j$-th derivative of a function $g$ is denoted by $g^{(j)}$. For vector $x \in \mathbb{R}^p$ and matrix $\mathbf{X} \in \mathbb{R}^{p \times p}$, $\|x\|_2$ and $\|\mathbf{X}\|_2$ denote the $\ell_2$ and spectral norms, respectively. $\mathcal{P}_\mathcal{C}$ is the Euclidean projection onto set $\mathcal{C}$, and $B_p(R) \subset \mathbb{R}^p$ is the ball of radius $R$. For random variables $x, y$, $d(x, y)$ and $\mathfrak{D}(x, y)$ denote probability metrics (to be explicitly defined later), measuring the distance between the distributions of $x$ and $y$.

## 2  Generalized Linear Models

Distribution of a random variable $y \in \mathbb{R}$ belongs to an exponential family with natural parameter $\eta \in \mathbb{R}$ if its density can be written of the form $f(y|\eta) = \exp\big(\eta y - \phi(\eta)\big)h(y)$, where $\phi$ is the *cumulant generating function* and $h$ is the *carrier density*. Let $y_1, y_2, ..., y_n$ be independent observations such that $\forall i \in [n], y_i \sim f(y_i|\eta_i)$. For $\eta = (\eta_1, ..., \eta_n)$, the joint likelihood is

$$f(y_1, y_2, ..., y_n | \eta) = \exp\left\{ \sum_{i=1}^{n} [y_i\eta_i - \phi(\eta_i)] \right\} \prod_{i=1}^{n} h(y_i).$$

We consider the problem of learning the maximum likelihood estimator in the above exponential family framework, where the vector $\eta \in \mathbb{R}^n$ is modeled through the linear relation,

$$\eta = \mathbf{X}\beta,$$

for some design matrix $\mathbf{X} \in \mathbb{R}^{n \times p}$ with rows $x_i \in \mathbb{R}^p$, and a coefficient vector $\beta \in \mathbb{R}^p$. This formulation is known as *Generalized Linear Models* (GLMs) in canonical form. The cumulant generating function $\phi$ determines the class of GLMs, i.e., for the ordinary least squares (OLS) $\phi(z) = z^2$ and for the logistic regression (LR) $\phi(z) = \log(1 + e^z)$.

Maximum likelihood estimation in the above formulation is equivalent to minimizing the negative log-likelihood function $\ell(\beta)$,

$$\ell(\beta) = \frac{1}{n} \sum_{i=1}^{n} [\phi(\langle x_i, \beta \rangle) - y_i \langle x_i, \beta \rangle], \tag{2.1}$$

where $\langle x, \beta \rangle$ is the inner product between the vectors $x$ and $\beta$. The relation to OLS and LR can be seen much easier by plugging in the corresponding $\phi(z)$ in Eq. (2.1). The gradient and the Hessian of $\ell(\beta)$ can be written as:

$$\nabla_\beta \ell(\beta) = \frac{1}{n} \sum_{i=1}^{n} \left[ \phi^{(1)}(\langle x_i, \beta \rangle) x_i - y_i x_i \right], \quad \nabla_\beta^2 \ell(\beta) = \frac{1}{n} \sum_{i=1}^{n} \phi^{(2)}(\langle x_i, \beta \rangle) x_i x_i^T. \tag{2.2}$$

For a sequence of scaling matrices $\{\mathbf{Q}^t\}_{t>0} \in \mathbb{R}^{p \times p}$, we consider iterations of the form

$$\hat{\beta}^{t+1} \leftarrow \hat{\beta}^t - \gamma_t \mathbf{Q}^t \nabla_\beta \ell(\hat{\beta}^t),$$

where $\gamma_t$ is the step size. The above iteration is our main focus, but with a new approach on how to compute the sequence of matrices $\{\mathbf{Q}^t\}_{t>0}$. We formulate the problem of finding a scalable $\mathbf{Q}^t$ as an estimation problem and use a *Stein*-type lemma that provides a computationally efficient update.

## 3  Newton-Stein Method

Classical Newton-Raphson update is generally used for training GLMs. However, its per-iteration cost makes it impractical for large-scale optimization. The main bottleneck is the computation of the Hessian matrix that requires $\mathcal{O}(np^2)$ flops which is prohibitive when $n \gg p \gg 1$. Numerous methods have been proposed to achieve NM's fast convergence rate while keeping the per-iteration cost manageable.

The task of constructing an approximate Hessian can be viewed as an estimation problem. Assuming that the rows of $\mathbf{X}$ are i.i.d. random vectors, the Hessian of GLMs with cumulant generating function $\phi$ has the following form

$$\left[\mathbf{Q}^t\right]^{-1} = \frac{1}{n} \sum_{i=1}^{n} x_i x_i^T \phi^{(2)}(\langle x_i, \beta \rangle) \approx \mathbb{E}[xx^T \phi^{(2)}(\langle x, \beta \rangle)].$$

We observe that $\left[\mathbf{Q}^t\right]^{-1}$ is just a sum of i.i.d. matrices. Hence, the true Hessian is nothing but a sample mean estimator to its expectation. Another natural estimator would be the sub-sampled Hessian method suggested by [Mar10, BCNN11, EM15]. Similarly, our goal is to propose an appropriate estimator that is also computationally efficient.

We use the following Stein-type lemma to derive an efficient estimator to the expectation of Hessian.

**Lemma 3.1** (Stein-type lemma). *Assume that $x \sim \mathsf{N}_p(0, \boldsymbol{\Sigma})$ and $\beta \in \mathbb{R}^p$ is a constant vector. Then for any function $f : \mathbb{R} \to \mathbb{R}$ that is twice "weakly" differentiable, we have*

$$\mathbb{E}[xx^T f(\langle x, \beta \rangle)] = \mathbb{E}[f(\langle x, \beta \rangle)]\boldsymbol{\Sigma} + \mathbb{E}[f^{(2)}(\langle x, \beta \rangle)]\boldsymbol{\Sigma}\beta\beta^T\boldsymbol{\Sigma}. \tag{3.1}$$

---

**Algorithm 1** Newton-Stein method

---

**Input:** $\hat{\beta}^0, r, \epsilon, \gamma$.

1. Set $t = 0$ and sub-sample a set of indices $S \subset [n]$ uniformly at random.

2. **Compute:** $\hat{\sigma}^2 = \lambda_{r+1}(\widehat{\boldsymbol{\Sigma}}_S),$ and $\quad \zeta_r(\widehat{\boldsymbol{\Sigma}}_S) = \hat{\sigma}^2 \mathbf{I} + \mathrm{argmin}_{\mathrm{rank}(M) = r} \left\| \widehat{\boldsymbol{\Sigma}}_S - \hat{\sigma}^2 \mathbf{I} - M \right\|_F.$

3. **while** $\left\| \hat{\beta}^{t+1} - \hat{\beta}^t \right\|_2 \leq \epsilon$ **do**

   $\hat{\mu}_2(\hat{\beta}^t) = \frac{1}{n} \sum_{i=1}^n \phi^{(2)}(\langle x_i, \hat{\beta}^t \rangle), \qquad \hat{\mu}_4(\hat{\beta}^t) = \frac{1}{n} \sum_{i=1}^n \phi^{(4)}(\langle x_i, \hat{\beta}^t \rangle),$

   $\mathbf{Q}^t = \frac{1}{\hat{\mu}_2(\hat{\beta}^t)} \left[ \zeta_r(\widehat{\boldsymbol{\Sigma}}_S)^{-1} - \frac{\hat{\beta}^t[\hat{\beta}^t]^T}{\hat{\mu}_2(\hat{\beta}^t)/\hat{\mu}_4(\hat{\beta}^t) + \langle \zeta_r(\widehat{\boldsymbol{\Sigma}}_S)\hat{\beta}^t, \hat{\beta}^t \rangle} \right],$

   $\hat{\beta}^{t+1} = \mathcal{P}_{B_p(R)} \left( \hat{\beta}^t - \gamma \mathbf{Q}^t \nabla_\beta \ell(\hat{\beta}^t) \right),$

   $t \leftarrow t + 1.$

4. **end while**

   **Output:** $\hat{\beta}^t$.

---

The proof of Lemma 3.1 is given in Appendix. The right hand side of Eq.(3.1) is a rank-1 update to the first term. Hence, its inverse can be computed with $\mathcal{O}(p^2)$ cost. Quantities that change at each iteration are the ones that depend on $\beta$, i.e.,

$$\mu_2(\beta) = \mathbb{E}[\phi^{(2)}(\langle x, \beta \rangle)] \quad \text{and} \quad \mu_4(\beta) = \mathbb{E}[\phi^{(4)}(\langle x, \beta \rangle)].$$

$\mu_2(\beta)$ and $\mu_4(\beta)$ are scalar quantities and can be estimated by their corresponding sample means $\hat{\mu}_2(\beta)$ and $\hat{\mu}_4(\beta)$ (explicitly defined at Step 3 of Algorithm 1), with only $\mathcal{O}(np)$ computation.

To complete the estimation task suggested by Eq. (3.1), we need an estimator for the covariance matrix $\boldsymbol{\Sigma}$. A natural estimator is the sample mean where, we only use a sub-sample $S \subset [n]$ so that the cost is reduced to $\mathcal{O}(|S|p^2)$ from $\mathcal{O}(np^2)$. Sub-sampling based sample mean estimator is denoted by $\widehat{\boldsymbol{\Sigma}}_S = \sum_{i \in S} x_i x_i^T / |S|$, which is widely used in large-scale problems [Ver10]. We highlight the fact that Lemma 3.1 replaces NM's $O(np^2)$ per-iteration cost with a one-time cost of $O(np^2)$. We further use sub-sampling to reduce this one-time cost to $O(|S|p^2)$.

In general, important curvature information is contained in the largest few spectral features. Following [EM15], we take the largest $r$ eigenvalues of the sub-sampled covariance estimator, setting rest of them to $(r + 1)$-th eigenvalue. This operation helps denoising and would require only $\mathcal{O}(rp^2)$ computation. Step 2 of Algorithm 1 performs this procedure.

Inverting the constructed Hessian estimator can make use of the low-rank structure several times. First, notice that the updates in Eq. (3.1) are based on rank-1 matrix additions. Hence, we can simply use a matrix inversion formula to derive an explicit equation (See $\mathbf{Q}^t$ in Step 3 of Algorithm 1). This formulation would impose another inverse operation on the covariance estimator. Since the covariance estimator is also based on rank-$r$ approximation, one can utilize the low-rank inversion formula again. We emphasize that this operation is performed once. Therefore, instead of NM's per-iteration cost of $\mathcal{O}(p^3)$ due to inversion, Newton-Stein method (NewSt) requires $\mathcal{O}(p^2)$ per-iteration and a one-time cost of $\mathcal{O}(rp^2)$. Assuming that NewSt and NM converge in $T_1$ and $T_2$ iterations respectively, the overall complexity of NewSt is $\mathcal{O}\left(npT_1 + p^2T_1 + (|S| + r)p^2\right) \approx \mathcal{O}\left(npT_1 + p^2T_1 + |S|p^2\right)$ whereas that of NM is $\mathcal{O}\left(np^2T_2 + p^3T_2\right)$.

Even though Proposition 3.1 assumes that the covariates are multivariate Gaussian random vectors, in Section 4, the only assumption we make on the covariates is that they have bounded support, which covers a wide class of random variables. The left plot of Figure 1 shows that the estimation is accurate for various distributions. This is a consequence of the fact that the proposed estimator in Eq. (3.1) relies on the distribution of $x$ only through inner products of the form $\langle x, v \rangle$, which in turn results in approximate normal distribution due to the central limit theorem when $p$ is sufficiently large. We will discuss this phenomenon in detail in Section 4.

The convergence rate of Newton-Stein method has two phases. Convergence starts quadratically and transitions into a linear rate when it gets close to the true minimizer. The phase transition behavior can be observed through the right plot in Figure 1. This is a consequence of the bound provided in Eq. (1.2), which is the main result of our theorems stated in Section 4.

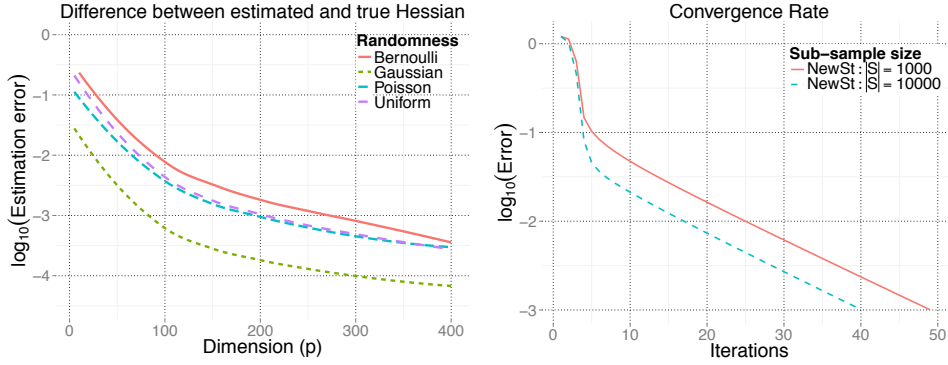

Figure 1: The left plot demonstrates the accuracy of proposed Hessian estimation over different distributions. Number of observations is set to be $n = \mathcal{O}(p \log(p))$. The right plot shows the phase transition in the convergence rate of Newton-Stein method (NewSt). Convergence starts with a quadratic rate and transitions into linear. Plots are obtained using *Covertype* dataset.

## 4 Theoretical results

We start this section by introducing the terms that will appear in the theorems. Then, we provide our technical results on uniformly bounded covariates. The proofs are provided in Appendix.

### 4.1 Preliminaries

Hessian estimation described in the previous section relies on a Gaussian approximation. For theoretical purposes, we use the following probability metric to quantify the gap between the distribution of $x_i$'s and that of a normal vector.

**Definition 1.** *Given a family of functions $\mathcal{H}$, and random vectors $x, y \in \mathbb{R}^p$, and any $h \in \mathcal{H}$, define*

$$d_{\mathcal{H}}(x, y) = \sup_{h \in \mathcal{H}} d_h(x, y) \quad \text{where} \quad d_h(x, y) = \left| \mathbb{E}\left[h(x)\right] - \mathbb{E}\left[h(y)\right] \right|.$$

Many probability metrics can be expressed as above by choosing a suitable function class $\mathcal{H}$. Examples include *Total Variation* (TV), *Kolmogorov* and *Wasserstein* metrics [GS02, CGS10]. Based on the second and fourth derivatives of cumulant generating function, we define the following classes:

$$\mathcal{H}_1 = \left\{ h(x) = \phi^{(2)}(\langle x, \beta \rangle) : \beta \in B_p(R) \right\}, \quad \mathcal{H}_2 = \left\{ h(x) = \phi^{(4)}(\langle x, \beta \rangle) : \beta \in B_p(R) \right\},$$

$$\mathcal{H}_3 = \left\{ h(x) = \langle v, x \rangle^2 \phi^{(2)}(\langle x, \beta \rangle) : \beta \in B_p(R), \|v\|_2 = 1 \right\},$$

where $B_p(R) \in \mathbb{R}^p$ is the ball of radius $R$. Exact calculation of such probability metrics are often difficult. The general approach is to upper bound the distance by a more intuitive metric. In our case, we observe that $d_{\mathcal{H}_j}(x, y)$ for $j = 1, 2, 3$, can be easily upper bounded by $d_{\text{TV}}(x, y)$ up to a scaling constant, when the covariates have bounded support.

We will further assume that the covariance matrix follows $r$-spiked model, i.e., $\Sigma = \sigma^2 I + \sum_{i=1}^{r} \theta_i u_i u_i^T$, which is commonly encountered in practice [BS06]. This simply means that the first $r$ eigenvalues of the covariance matrix are large and the rest are small and equal to each other. Large eigenvalues of $\Sigma$ correspond to the signal part and small ones (denoted by $\sigma^2$) can be considered as the noise component.

### 4.2 Composite convergence rate

We have the following per-step bound for the iterates generated by the Newton-Stein method, when the covariates are supported on a $p$-dimensional ball.

**Theorem 4.1.** *Assume that the covariates $x_1, x_2, ..., x_n$ are i.i.d. random vectors supported on a ball of radius $\sqrt{K}$ with*

$$\mathbb{E}[x_i] = 0 \quad \text{and} \quad \mathbb{E}\left[x_i x_i^T\right] = \Sigma,$$

*where $\Sigma$ follows the $r$-spiked model. Further assume that the cumulant generating function $\phi$ has bounded 2nd-5th derivatives and that $R$ is the radius of the projection $\mathcal{P}_{B_p(R)}$. For $\left\{\hat{\beta}^t\right\}_{t>0}$ given*

*by the Newton-Stein method for $\gamma = 1$, define the event*

$$\mathcal{E} = \left\{ \left| \mu_2(\hat{\beta}^t) + \mu_4(\hat{\beta}^t)\langle \boldsymbol{\Sigma}\hat{\beta}^t, \hat{\beta}^t \rangle \right| > \xi, \ \beta_* \in B_p(R) \right\} \tag{4.1}$$

*for some positive constant $\xi$, and the optimal value $\beta_*$. If $n, |S|$ and $p$ are sufficiently large, then there exist constants $c, c_1, c_2$ and $\kappa$ depending on the radii $K, R, \mathbb{P}(\mathcal{E})$ and the bounds on $|\phi^{(2)}|$ and $|\phi^{(4)}|$ such that conditioned on the event $\mathcal{E}$, with probability at least $1 - c/p^2$, we have*

$$\left\| \hat{\beta}^{t+1} - \beta_* \right\|_2 \leq \tau_1 \left\| \hat{\beta}^t - \beta_* \right\|_2 + \tau_2 \left\| \hat{\beta}^t - \beta_* \right\|_2^2, \tag{4.2}$$

*where the coefficients $\tau_1$ and $\tau_2$ are deterministic constants defined as*

$$\tau_1 = \kappa \mathfrak{D}(x, z) + c_1 \kappa \sqrt{\frac{p}{\min\left\{ p/\log(p)|S|, n/\log(n) \right\}}}, \qquad \tau_2 = c_2 \kappa,$$

*and $\mathfrak{D}(x, z)$ is defined as*

$$\mathfrak{D}(x, z) = \|\boldsymbol{\Sigma}\|_2 \, d_{\mathcal{H}_1}(x, z) + \|\boldsymbol{\Sigma}\|_2^2 R^2 \, d_{\mathcal{H}_2}(x, z) + d_{\mathcal{H}_3}(x, z), \tag{4.3}$$

*for a multivariate Gaussian random variable $z$ with the same mean and covariance as $x_i$'s.*

The bound in Eq. (4.2) holds with high probability, and the coefficients $\tau_1$ and $\tau_2$ are deterministic constants which will describe the convergence behavior of the Newton-Stein method. Observe that the coefficient $\tau_1$ is sum of two terms: $\mathfrak{D}(x, z)$ measures how accurate the Hessian estimation is, and the second term depends on the sub-sample size and the data dimensions.

Theorem 4.1 shows that the convergence of Newton-Stein method can be upper bounded by a compositely converging sequence, that is, the squared term will dominate at first giving a quadratic rate, then the convergence will transition into a linear phase as the iterate gets close to the optimal value. The coefficients $\tau_1$ and $\tau_2$ govern the linear and quadratic terms, respectively. The effect of sub-sampling appears in the coefficient of linear term. In theory, there is a threshold for the sub-sampling size $|S|$, namely $\mathcal{O}(n/\log(n))$, beyond which further sub-sampling has no effect. The transition point between the quadratic and the linear phases is determined by the sub-sampling size and the properties of the data. The phase transition can be observed through the right plot in Figure 1. Using the above theorem, we state the following corollary.

**Corollary 4.2.** *Assume that the assumptions of Theorem 4.1 hold. For a constant $\delta \geq \mathbb{P}\left(\mathcal{E}^C\right)$, a tolerance $\epsilon$ satisfying*

$$\epsilon \geq 20R \left\{ c/p^2 + \delta \right\},$$

*and for an iterate satisfying $\mathbb{E}\left[\|\hat{\beta}^t - \beta_*\|_2\right] > \epsilon$, the iterates of Newton-Stein method will satisfy,*

$$\mathbb{E}\left[ \|\hat{\beta}^{t+1} - \beta_*\|_2 \right] \leq \tilde{\tau}_1 \mathbb{E}\left[ \|\hat{\beta}^t - \beta_*\|_2 \right] + \tau_2 \mathbb{E}\left[ \|\hat{\beta}^t - \beta_*\|_2^2 \right],$$

*where $\tilde{\tau}_1 = \tau_1 + 0.1$ and , $\tau_1, \tau_2$ are as in Theorem 4.1.*

The bound stated in the above corollary is an analogue of composite convergence (given in Eq. (4.2)) in expectation. Note that our results make strong assumptions on the derivatives of the cumulant generating function $\phi$. We emphasize that these assumptions are valid for linear and logistic regressions. An example that does not fit in our scheme is *Poisson regression* with $\phi(z) = e^z$. However, we observed empirically that the algorithm still provides significant improvement. The following theorem states a sufficient condition for the convergence of composite sequence.

**Theorem 4.3.** *Let $\{\hat{\beta}^t\}_{t\geq 0}$ be a compositely converging sequence with convergence coefficients $\tau_1$ and $\tau_2$ as in Eq. (4.2) to the minimizer $\beta_*$. Let the starting point satisfy $\left\|\hat{\beta}^0 - \beta_*\right\|_2 = \vartheta < (1 - \tau_1)/\tau_2$ and define $\Xi = \left( \frac{\tau_1 \vartheta}{1 - \tau_2 \vartheta}, \vartheta \right)$. Then the sequence of $\ell_2$-distances converges to 0. Further, the number of iterations to reach a tolerance of $\epsilon$ can be upper bounded by $\inf_{\xi \in \Xi} \mathcal{J}(\xi)$, where*

$$\mathcal{J}(\xi) = \log_2\left( \frac{\log\left(\delta\left(\tau_1/\xi + \tau_2\right)\right)}{\log\left(\tau_1/\xi + \tau_2\right)\vartheta} \right) + \frac{\log(\epsilon/\xi)}{\log(\tau_1 + \tau_2\xi)}. \tag{4.4}$$

Above theorem gives an upper bound on the number of iterations until reaching a tolerance of $\epsilon$. The first and second terms on the right hand side of Eq. (4.4) stem from the quadratic and linear phases, respectively.

### 4.3 Algorithm parameters

NewSt takes three input parameters and for those, we suggest near-optimal choices based on our theoretical results.

- **Sub-sample size:** NewSt uses a subset of indices to approximate the covariance matrix $\Sigma$. Corollary 5.50 of [Ver10] proves that a sample size of $\mathcal{O}(p)$ is sufficient for sub-gaussian covariates and that of $\mathcal{O}(p\log(p))$ is sufficient for arbitrary distributions supported in some ball to estimate a covariance matrix by its sample mean estimator. In the regime we consider, $n \gg p$, we suggest to use a sample size of $|S| = \mathcal{O}(p\log(p))$.
- **Rank:** Many methods have been suggested to improve the estimation of covariance matrix and almost all of them rely on the concept of *shrinkage* [CCS10, DGJ13]. Eigenvalue thresholding can be considered as a shrinkage operation which will retain only the important second order information [EM15]. Choosing the rank threshold $r$ can be simply done on the sample mean estimator of $\Sigma$. After obtaining the sub-sampled estimate of the mean, one can either plot the spectrum and choose manually or use a technique from [DG13].
- **Step size:** Step size choices of NewSt are quite similar to Newton's method (i.e., See [BV04]). The main difference comes from the eigenvalue thresholding. If the data follows the $r$-spiked model, the optimal step size will be close to 1 if there is no sub-sampling. However, due to fluctuations resulting from sub-sampling, we suggest the following step size choice for NewSt:

$$\gamma = \frac{2}{1 + \frac{\hat{\sigma}^2 - \mathcal{O}(\sqrt{p/|S|})}{\hat{\sigma}^2}}. \tag{4.5}$$

  In general, this formula yields a step size greater than 1, which is due to rank thresholding, providing faster convergence. See [EM15] for a detailed discussion.

## 5 Experiments

In this section, we validate the performance of NewSt through extensive numerical studies. We experimented on two commonly used GLM optimization problems, namely, *Logistic Regression* (LR) and *Linear Regression* (OLS). LR minimizes Eq. (2.1) for the logistic function $\phi(z) = \log(1 + e^z)$, whereas OLS minimizes the same equation for $\phi(z) = z^2$. In the following, we briefly describe the algorithms that are used in the experiments:

- **Newton's Method** (NM) uses the inverse Hessian evaluated at the current iterate, and may achieve quadratic convergence. NM steps require $\mathcal{O}(np^2 + p^3)$ computation which makes it impractical for large-scale datasets.
- **Broyden-Fletcher-Goldfarb-Shanno** (BFGS) forms a curvature matrix by cultivating the information from the iterates and the gradients at each iteration. Under certain assumptions, the convergence rate is locally super-linear and the per-iteration cost is comparable to that of first order methods.
- **Limited Memory BFGS** (L-BFGS) is similar to BFGS, and uses only the recent few iterates to construct the curvature matrix, gaining significant performance in terms of memory.
- **Gradient Descent** (GD) update is proportional to the negative of the full gradient evaluated at the current iterate. Under smoothness assumptions, GD achieves a linear convergence rate, with $\mathcal{O}(np)$ per-iteration cost.
- **Accelerated Gradient Descent** (AGD) is proposed by Nesterov [Nes83], which improves over the gradient descent by using a momentum term. Performance of AGD strongly depends of the smoothness of the function.

For all the algorithms, we use a constant step size that provides the fastest convergence. Sub-sample size, rank and the constant step size for NewSt is selected by following the guidelines in Section 4.3.

We experimented over two real, two synthetic datasets which are summarized in Table 1. Synthetic data are generated through a multivariate Gaussian distribution and data dimensions are chosen so that Newton's method still does well. The experimental results are summarized in Figure 2. We observe that NewSt provides a significant improvement over the classical techniques. The methods that come closer to NewSt is Newton's method for moderate $n$ and $p$ and BFGS when $n$ is large.

Observe that the convergence rate of NewSt has a clear phase transition point. As argued earlier, this point depends on various factors including sub-sampling size $|S|$ and data dimensions $n, p$, the

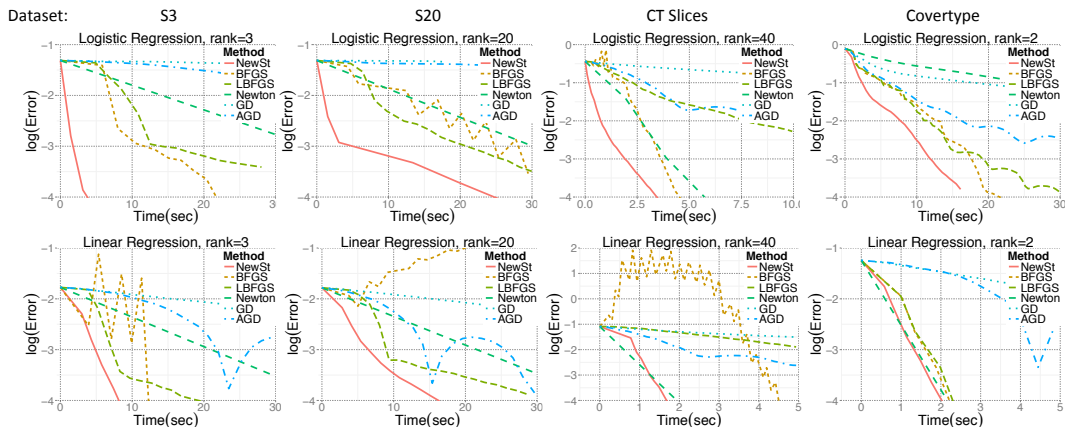

Figure 2: Performance of various optimization methods on different datasets. Red straight line represents the proposed method NewSt. Algorithm parameters including the rank threshold is selected by the guidelines described in Section 4.3.

rank threshold $r$ and structure of the covariance matrix. The prediction of the phase transition point is an interesting line of research, which would allow further tuning of algorithm parameters.

The optimal step-size for NewSt will typically be larger than 1 which is mainly due to the eigenvalue thresholding operation. This feature is desirable if one is able to obtain a large step-size that provides convergence. In such cases, the convergence is likely to be faster, yet more unstable compared to the smaller step size choices. We observed that similar to other second order algorithms, NewSt is susceptible to the step size selection. If the data is not well-conditioned, and the sub-sample size is not sufficiently large, algorithm might have poor performance. This is mainly because the sub-sampling operation is performed only once at the beginning. Therefore, it might be good in practice to sub-sample once in every few iterations.

| Dataset | $n$ | $p$ | Reference, UCI repo [Lic13] |
|---|---|---|---|
| CT slices | 53500 | 386 | [GKS$^+$11] |
| Covertype | 581012 | 54 | [BD99] |
| S3 | 500000 | 300 | 3-spiked model, [DGJ13] |
| S20 | 500000 | 300 | 20-spiked model, [DGJ13] |

Table 1: Datasets used in the experiments.

# 6 Discussion

In this paper, we proposed an efficient algorithm for training GLMs. We call our algorithm Newton-Stein method (NewSt) as it takes a Newton update at each iteration relying on a Stein-type lemma. The algorithm requires a one time $\mathcal{O}(|S|p^2)$ cost to estimate the covariance structure and $\mathcal{O}(np)$ per-iteration cost to form the update equations. We observe that the convergence of NewSt has a phase transition from quadratic rate to linear. This observation is justified theoretically along with several other guarantees for covariates with bounded support, such as per-step bounds, conditions for convergence, etc. Parameter selection guidelines of NewSt are based on our theoretical results. Our experiments show that NewSt provides high performance in GLM optimization.

Relaxing some of the theoretical constraints is an interesting line of research. In particular, bounded support assumption as well as strong constraints on the cumulant generating functions might be loosened. Another interesting direction is to determine when the phase transition point occurs, which would provide a better understanding of the effects of sub-sampling and rank thresholding.

## Acknowledgements

The author is grateful to Mohsen Bayati and Andrea Montanari for stimulating conversations on the topic of this work. The author would like to thank Bhaswar B. Bhattacharya and Qingyuan Zhao for carefully reading this article and providing valuable feedback.

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
