[Supplementary Material]



# Newton-Stein Method for GLMs

We provide all technical details in the Appendix. Section A provides the derivation of the Stein-type lemma that lies at the core of our algorithm. Section B includes useful concentration results that will be used in the proof of main theorem. In Section C, we prove the theorems that appeared in the main text. In the last section, we state several auxiliary lemmas that are used throughout the proofs.

## A    Proof of Stein-type lemma

*Proof of Lemma 3.1.* The proof will follow from integration by parts over multivariate variables. Let $g(x)$ be the density of $x$, i.e.,

$$g(x) = (2\pi)^{-p/2} |\boldsymbol{\Sigma}|^{-1/2} \exp\left\{ -\frac{1}{2}\langle \boldsymbol{\Sigma}^{-1}x, x \rangle \right\},$$

and $xg(x)dx = -\boldsymbol{\Sigma} dg(x)$. We write

$$\mathbb{E}[xx^T f(\langle x, \beta\rangle)] = \int xx^T f(\langle x, \beta\rangle) g(x) dx,$$

$$= \boldsymbol{\Sigma} \int -f(\langle x, \beta\rangle) dg(x) x^T,$$

$$= \boldsymbol{\Sigma} \left\{ \int f(\langle x, \beta\rangle) g(x) dx + \int \beta x^T f^{(1)}(\langle x, \beta\rangle) g(x) dx \right\},$$

$$= \boldsymbol{\Sigma} \left\{ \mathbb{E}[f(\langle x, \beta\rangle)] + \int \beta\beta^T f^{(2)}(\langle x, \beta\rangle) g(x) dx \boldsymbol{\Sigma} \right\},$$

$$= \boldsymbol{\Sigma} \left\{ \mathbb{E}[f(\langle x, \beta\rangle)] + \beta\beta^T \mathbb{E}[f^{(2)}(\langle x, \beta\rangle)] \boldsymbol{\Sigma} \right\},$$

$$= \mathbb{E}[f(\langle x, \beta\rangle)]\boldsymbol{\Sigma} + \mathbb{E}[f^{(2)}(\langle x, \beta\rangle)]\boldsymbol{\Sigma}\beta\beta^T \boldsymbol{\Sigma},$$

which is the desired result. $\qquad\qquad\square$

## B    Preliminary concentration inequalities

In this section, we provide several results that will be useful in the proof of main theorem. We start with some simple definitions on a special class of random variables.

**Definition 2** (Sub-gaussian)**.** *For a given constant $K$, a random variable $x \in \mathbb{R}$ is called* sub-gaussian *if it satisfies*

$$\mathbb{E}[|x|^m]^{1/m} \leq K\sqrt{m}, \quad m \geq 1.$$

*Smallest such $K$ is the sub-gaussian norm of $x$ and it is denoted by $\|x\|_{\psi_2}$. Similarly, a random vector $y \in \mathbb{R}^p$ is a* sub-gaussian vector *if there exists a constant $K'$ such that*

$$\sup_{v \in S^{p-1}} \|\langle y, v\rangle\|_{\psi_2} \leq K'.$$

**Definition 3** (Sub-exponential)**.** *For a given constant $K$, a random variable $x \in \mathbb{R}$ is called* sub-exponential *if it satisfies*

$$\mathbb{E}[|x|^m]^{1/m} \leq Km, \quad m \geq 1,$$

*Smallest such $K$ is the sub-exponential norm of $x$ and it is denoted by $\|x\|_{\psi_1}$. Similarly, a random vector $y \in \mathbb{R}^p$ is a* sub-exponential vector *if there exists a constant $K'$ such that*

$$\sup_{v \in S^{p-1}} \|\langle y, v\rangle\|_{\psi_1} \leq K'.$$

We state the following Lemma from [Ver10] (i.e., see Theorem 5.44 and Corollary 5.52):

**Lemma B.1.** *Let $S$ be an index set and $x_i \in \mathbb{R}^p$ for $i \in S$ be i.i.d. random vectors with*

$$\mathbb{E}[x_i] = 0, \qquad \mathbb{E}[x_i x_i^T] = \mathbf{\Sigma}, \qquad \|x_i\|_2 \le \sqrt{K} \text{ a.s.}$$

*for a covariance matrix $\mathbf{\Sigma}$ and a constant $K$. For a small $\epsilon$, if the sample size satisfies $|S| > CK^2 \log(p)/\epsilon^2$, then with probability $1 - 1/p^2$, we have*

$$\left\| \widehat{\mathbf{\Sigma}}_S - \mathbf{\Sigma} \right\|_2 \le \epsilon.$$

**Remark 1.** *The above lemma suggests that if the sample size is sufficiently large, i.e., $|S| = \mathcal{O}(K^2 \log(p))$, we can estimate the true covariance matrix quite well. In particular, with probability $1 - 1/p^2$, we obtain*

$$\left\| \widehat{\mathbf{\Sigma}}_S - \mathbf{\Sigma} \right\|_2 \le c\sqrt{\frac{\log(p)}{|S|}},$$

*where $c = K\sqrt{C}$.*

The following lemma will be helpful to show a similar concentration result for the random matrix $\zeta_r(\widehat{\mathbf{\Sigma}}_S)$:

**Lemma B.2.** *Let the assumptions in Lemma B.1 hold. Further, assume that $\mathbf{\Sigma}$ follows $r$-spiked model. If $|S|$ is sufficiently large, for $c = 2K\sqrt{C}$, with probability $1 - 1/p^2$, we have*

$$\left\| \zeta_r(\widehat{\mathbf{\Sigma}}_S) - \widehat{\mathbf{\Sigma}}_S \right\|_2 \le c\sqrt{\frac{\log(p)}{|S|}},$$

*where $C$ is an absolute constant.*

*Proof.* By the Weyl's inequality for the eigenvalues, we have

$$\left\| \zeta_r(\widehat{\mathbf{\Sigma}}_S) - \widehat{\mathbf{\Sigma}}_S \right\|_2 = \lambda_{r+1}(\widehat{\mathbf{\Sigma}}_S) - \lambda_p(\widehat{\mathbf{\Sigma}}_S) \le 2\|\widehat{\mathbf{\Sigma}}_S - \mathbf{\Sigma}\|_2.$$

Hence the result follows from the previous lemma. $\qquad\qquad\square$

Note that the same bound also applies to $\left\| \zeta_r(\widehat{\mathbf{\Sigma}}_S) - \mathbf{\Sigma} \right\|_2$. Lemmas B.1 and B.2 are standard concentration results for the random matrices with i.i.d. rows. The following tool will be used to obtain upper bounds for the empirical processes.

**Definition 4.** *On a metric space $(X, d)$, for $\epsilon > 0$, $T_\epsilon \subset X$ is called an $\epsilon$-net over $X$ if $\forall x \in X$, $\exists t \in T_\epsilon$ such that $d(x, t) \le \epsilon$.*

Preliminary tools presented in this section will be used to obtain the main concentration results in Section C.

## C   Main lemmas

### C.1   Concentration of covariates with bounded support

**Lemma C.1.** *Let $x_i \in \mathbb{R}^p$, for $i = 1, 2, ..., n$, be i.i.d. random vectors supported on a ball of radius $\sqrt{K}$, with mean 0, and covariance matrix $\mathbf{\Sigma}$. Also let $f : \mathbb{R} \to \mathbb{R}$ be a uniformly bounded function such that for some $B > 0$, we have $\|f\|_\infty < B$ and $f$ is Lipschitz continuous with constant $L$. Then, there exist constants $c_1, c_2, c_3$ such that*

$$\mathbb{P}\left( \sup_{\beta \in B_n(R)} \left| \frac{1}{n}\sum_{i=1}^{n} f(\langle x_i, \beta \rangle) - \mathbb{E}[f(\langle x, \beta \rangle)] \right| > c_1\sqrt{\frac{p\log(n)}{n}} \right) \le c_2 e^{-c_3 p},$$

*where the constants depend only on the bound $B$ and radii $R$ and $\sqrt{K}$.*

*Proof of Lemma C.1.* We start by using the Lipschitz property of the function $f$, i.e., $\forall \beta, \beta' \in B_p(R)$,

$$\|f(\langle x, \beta \rangle) - f(\langle x, \beta' \rangle)\|_2 \leq L\|x\|_2 \|\beta - \beta'\|_2,$$
$$\leq L\sqrt{K}\|\beta - \beta'\|_2.$$

Now let $T_\Delta$ be a $\Delta$-net over $B_p(R)$. Then $\forall \beta \in B_p(R)$, $\exists \beta' \in T_\Delta$ such that right hand side of the above inequality is smaller than $\Delta L\sqrt{K}$. Then, we can write

$$\left| \frac{1}{n} \sum_{i=1}^n f(\langle x_i, \beta \rangle) - \mathbb{E}[f(\langle x, \beta \rangle)] \right| \leq \left| \frac{1}{n} \sum_{i=1}^n f(\langle x_i, \beta' \rangle) - \mathbb{E}[f(\langle x, \beta' \rangle)] \right| + 2\Delta L\sqrt{K}. \quad \text{(C.1)}$$

By choosing

$$\Delta = \frac{\epsilon}{4L\sqrt{K}},$$

and taking supremum over the corresponding $\beta$ sets on both sides, we obtain the following inequality

$$\sup_{\beta \in B_n(R)} \left| \frac{1}{n} \sum_{i=1}^n f(\langle x_i, \beta \rangle) - \mathbb{E}[f(\langle x, \beta \rangle)] \right| \leq \max_{\beta \in T_\Delta} \left| \frac{1}{n} \sum_{i=1}^n f(\langle x_i, \beta \rangle) - \mathbb{E}[f(\langle x, \beta \rangle)] \right| + \frac{\epsilon}{2}.$$

Now, since we have $\|f\|_\infty \leq B$ and for a fixed $\beta$ and $i = 1, 2, ..., n$, the random variables $f(\langle x_i, \beta \rangle)$ are i.i.d., by the Hoeffding's concentration inequality, we have

$$\mathbb{P}\left( \left| \frac{1}{n} \sum_{i=1}^n f(\langle x_i, \beta \rangle) - \mathbb{E}[f(\langle x, \beta \rangle)] \right| > \epsilon/2 \right) \leq 2\exp\left( -\frac{n\epsilon^2}{2B^2} \right).$$

Combining Eq. (C.1), the above result together with the union bound, we easily obtain

$$\mathbb{P}\left( \sup_{\beta \in B_n(R)} \left| \frac{1}{n} \sum_{i=1}^n f(\langle x_i, \beta \rangle) - \mathbb{E}[f(\langle x, \beta \rangle)] \right| > \epsilon \right) \leq \mathbb{P}\left( \max_{\beta \in T_\Delta} \left| \frac{1}{n} \sum_{i=1}^n f(\langle x_i, \beta \rangle) - \mathbb{E}[f(\langle x, \beta \rangle)] \right| > \epsilon/2 \right)$$
$$\leq 2|T_\Delta| \exp\left( -\frac{n\epsilon^2}{2B^2} \right),$$

where $\Delta = \epsilon/4L\sqrt{K}$.

Next, we apply Lemma E.2 and obtain that

$$|T_\Delta| \leq \left( \frac{R\sqrt{p}}{\Delta} \right)^p = \left( \frac{R\sqrt{p}}{\epsilon/4L\sqrt{K}} \right)^p.$$

We require that the bound on the probability gets an exponential decay with rate $\mathcal{O}(p)$. Using Lemma E.3 with $a = 2B^2 p/n$ and $b = 4LR\sqrt{K}p$, we obtain that $\epsilon$ should be

$$\epsilon = \sqrt{\frac{B^2 p}{n} \log\left( \frac{16L^2 R^2 Kn}{B^2} \right)} = \mathcal{O}\left( \sqrt{\frac{p\log(n)}{n}} \right),$$

which completes the proof. $\qquad \square$

In the following, we state the concentration results on functions of the form

$$x \to f(\langle x, \beta \rangle)\langle x, v \rangle^2.$$

Functions of this type form the summands of the Hessian matrix in GLMs.

**Lemma C.2.** *Let $x_i \in \mathbb{R}^p$, for $i = 1, ..., n$, be i.i.d. random vectors supported on a ball of radius $\sqrt{K}$, with mean 0, covariance matrix $\Sigma$. Also let $f : \mathbb{R} \to \mathbb{R}$ be a uniformly bounded function such that for some $B > 0$, we have $\|f\|_\infty < B$ and $f$ is Lipschitz continuous with constant L. Then, for*

$v \in S^{p-1}$, *there exist constants $c_1, c_2, c_3$ such that*

$$\mathbb{P}\left(\sup_{\beta \in B_p(R)} \left| \frac{1}{n} \sum_{i=1}^{n} f(\langle x_i, \beta \rangle)\langle x_i, v \rangle^2 - \mathbb{E}[f(\langle x, \beta \rangle)\langle x, v \rangle^2] \right| > c_1 \sqrt{\frac{p \log(n)}{n}} \right) \leq c_2 e^{-c_3 p},$$

*where the constants depend only on the bound $B$ and radii $R$ and $\sqrt{K}$.*

*Proof of Lemma C.2.* As in the proof of Lemma C.1, we start by using the Lipschitz property of the function $f$, i.e., $\forall \beta, \beta' \in B_p(R)$,

$$\|f(\langle x, \beta \rangle)\langle x, v \rangle^2 - f(\langle x, \beta' \rangle)\langle x, v \rangle^2\|_2 \leq L\|x\|_2^3 \|\beta - \beta'\|_2,$$
$$\leq LK^{1.5}\|\beta - \beta'\|_2.$$

For a net $T_\Delta$, $\forall \beta \in B_p(R)$, $\exists \beta' \in T_\Delta$ such that right hand side of the above inequality is smaller than $\Delta L \sqrt{K}$. Then, we can write

$$\left| \frac{1}{n} \sum_{i=1}^{n} f(\langle x_i, \beta \rangle)\langle x_i, v \rangle^2 - \mathbb{E}[f(\langle x, \beta \rangle)\langle x, v \rangle^2] \right| \leq \left| \frac{1}{n} \sum_{i=1}^{n} f(\langle x_i, \beta' \rangle)\langle x_i, v \rangle^2 - \mathbb{E}[f(\langle x, \beta' \rangle)\langle x, v \rangle^2] \right|$$
$$+ 2\Delta L K^{1.5}. \tag{C.2}$$

This time, we choose

$$\Delta = \frac{\epsilon}{4LK^{1.5}},$$

and take the supremum over the corresponding feasible $\beta$-sets on both sides,

$$\sup_{\beta \in B_n(R)} \left| \frac{1}{n} \sum_{i=1}^{n} f(\langle x_i, \beta \rangle)\langle x_i, v \rangle^2 - \mathbb{E}[f(\langle x, \beta \rangle)\langle x, v \rangle^2] \right|$$
$$\leq \max_{\beta \in T_\Delta} \left| \frac{1}{n} \sum_{i=1}^{n} f(\langle x_i, \beta \rangle)\langle x_i, v \rangle^2 - \mathbb{E}[f(\langle x, \beta \rangle)\langle x, v \rangle^2] \right| + \frac{\epsilon}{2}.$$

Now, since we have $\|f\|_\infty \leq B$ and for fixed $\beta$ and $v$, $i = 1, 2, ..., n$, $f(\langle x_i, \beta \rangle)\langle x_i, v \rangle^2$ are i.i.d. random variables. By the Hoeffding's concentration inequality, we have

$$\mathbb{P}\left( \left| \frac{1}{n} \sum_{i=1}^{n} f(\langle x_i, \beta \rangle)\langle x_i, v \rangle^2 - \mathbb{E}[f(\langle x, \beta \rangle)\langle x, v \rangle^2] \right| > \epsilon/2 \right) \leq 2 \exp\left( -\frac{n\epsilon^2}{2B^2K^2} \right).$$

Using Eq. (C.2) and the above result combined with the union bound, we easily obtain

$$\mathbb{P}\left( \sup_{\beta \in B_n(R)} \left| \frac{1}{n} \sum_{i=1}^{n} f(\langle x_i, \beta \rangle)\langle x_i, v \rangle^2 - \mathbb{E}[f(\langle x, \beta \rangle)\langle x, v \rangle^2] \right| > \epsilon \right)$$
$$\leq \mathbb{P}\left( \max_{\beta \in T_\Delta} \left| \frac{1}{n} \sum_{i=1}^{n} f(\langle x_i, \beta \rangle)\langle x_i, v \rangle^2 - \mathbb{E}[f(\langle x, \beta \rangle)\langle x, v \rangle^2] \right| > \epsilon/2 \right)$$
$$\leq 2|T_\Delta| \exp\left( -\frac{n\epsilon^2}{2B^2K^2} \right),$$

where $\Delta = \epsilon/4LK^{1.5}$. Using Lemma E.2, we have

$$|T_\Delta| \leq \left( \frac{R\sqrt{p}}{\Delta} \right)^p = \left( \frac{R\sqrt{p}}{\epsilon/4LK^{1.5}} \right)^p.$$

As before, we require that the right hand side of above inequality gets a decay with rate $\mathcal{O}(p)$. Using Lemma E.3 with $a = 2B^2K^2p/n$ and $b = 4LRK^{1.5}\sqrt{p}$, we obtain that $\epsilon$ should be

$$\epsilon = \sqrt{\frac{B^2K^2p}{n}\log\left(\frac{16L^2R^2K^3n}{B^2}\right)} = \mathcal{O}\left(\sqrt{\frac{p\log(n)}{n}}\right),$$

which completes the proof. $\qquad\square$

## D  Proof of main theorem

The proof will follow from the concentration results derived in previous sections. Our matrix concentration results are based on the covering net argument provided in [Ver10]. Similar results can also be obtained through different techniques such as *chaining* [DE15].

On the set $\mathcal{E}$, we have

$$\hat{\beta}^t - \beta_* - \gamma\mathbf{Q}^t\nabla_\beta\ell(\hat{\beta}^t) = \hat{\beta}^t - \beta_* - \gamma\mathbf{Q}^t\int_0^1 \nabla_\beta^2\ell(\beta_* + \xi(\hat{\beta}^t - \beta_*))d\xi(\hat{\beta}^t - \beta_*),$$

$$= \left(I - \gamma\mathbf{Q}^t\int_0^1 \nabla_\beta^2\ell(\beta_* + \xi(\hat{\beta}^t - \beta_*))d\xi\right)(\hat{\beta}^t - \beta_*).$$

Since the projection $\mathcal{P}_{B_p(R)}$ in step 3 of NewSt can only decrease the $\ell_2$ distance, we obtain

$$\|\hat{\beta}^{t+1} - \beta_*\|_2 \le \left\|I - \gamma\mathbf{Q}^t\int_0^1 \nabla_\beta^2\ell(\beta_* + \xi(\hat{\beta}^t - \beta_*))d\xi\right\|_2 \|\hat{\beta}^t - \beta_*\|_2. \qquad (D.1)$$

The governing term (with $\gamma = 1$) that determines the convergence rate can be bounded as

$$\left\|I - \mathbf{Q}^t\int_0^1 \nabla_\beta^2\ell(\beta_* + \xi(\hat{\beta}^t - \beta_*))d\xi\right\|_2 \le \left\|[\mathbf{Q}^t]^{-1} - \int_0^1 \nabla_\beta^2\ell(\beta_* + \xi(\hat{\beta}^t - \beta_*))d\xi\right\|_2 \|\mathbf{Q}^t\|_2.$$

We define the following,

$$\mathfrak{E}(\beta) = \mathbb{E}[\phi^{(2)}(\langle x, \beta\rangle)]\mathbf{\Sigma} + \mathbb{E}\left[\phi^{(4)}(\langle x, \beta\rangle)\right]\mathbf{\Sigma}\beta\beta^T\mathbf{\Sigma}$$

Note that for a function $f$, $\mathbb{E}[f(\langle x, \beta\rangle)] = h(\beta)$ is a function of $\beta$. With a slight abuse of notation, we write $\mathbb{E}[f(\langle x, \hat{\beta}\rangle)] = h(\hat{\beta})$ as a random variable. We have

$$\left\|[\mathbf{Q}^t]^{-1} - \int_0^1 \nabla_\beta^2\ell(\beta_* + \xi(\hat{\beta}^t - \beta_*))d\xi\right\|_2 \le \left\|[\mathbf{Q}^t]^{-1} - \mathfrak{E}(\hat{\beta}^t)\right\|_2$$

$$+ \left\|\mathbb{E}[xx^T\phi^{(2)}(\langle x, \hat{\beta}^t\rangle)] - \mathfrak{E}(\hat{\beta}^t)\right\|_2$$

$$+ \left\|\int_0^1 \nabla_\beta^2\ell(\beta_* + \xi(\hat{\beta}^t - \beta_*))d\xi - \mathbb{E}\left[xx^T\int_0^1 \phi^{(2)}(\langle x, \beta_* + \xi(\hat{\beta}^t - \beta_*)\rangle)d\xi\right]\right\|_2$$

$$+ \left\|\mathbb{E}[xx^T\phi^{(2)}(\langle x, \hat{\beta}^t\rangle)] - \mathbb{E}\left[xx^T\int_0^1 \phi^{(2)}(\langle x, \beta_* + \xi(\hat{\beta}^t - \beta_*)\rangle)d\xi\right]\right\|_2.$$

For the first term on the right hand side, we state the following lemma.

**Lemma D.1.** *There exist constants $c, C$ such that, with probability at least $1 - c/p^2$,*

$$\left\|[\mathbf{Q}^t]^{-1} - \mathfrak{E}(\hat{\beta}^t)\right\|_2 \le C\sqrt{\frac{p}{\min\left\{|S|p/\log(p), n/\log(n)\right\}}},$$

*where the constants depend on $K$, $B$ and the radius $R$.*

*Proof of Lemma D.1.* Using a uniform bound on the feasible set, we write

$$\left\| [\mathbf{Q}^t]^{-1} - \mathfrak{E}(\hat{\beta}^t) \right\|_2$$
$$\leq \sup_{\beta \in B_p(R)} \left\| \hat{\mu}_2(\beta) \zeta_r(\widehat{\mathbf{\Sigma}}_S) + \hat{\mu}_4(\beta) \zeta_r(\widehat{\mathbf{\Sigma}}_S) \beta (\zeta_r(\widehat{\mathbf{\Sigma}}_S) \beta)^T - \mathbb{E}[\phi^{(2)}(\langle x, \beta \rangle)] \mathbf{\Sigma} - \mathbb{E}[\phi^{(4)}(\langle x, \beta \rangle)] \mathbf{\Sigma} \beta \beta^T \mathbf{\Sigma} \right\|_2.$$

We will find an upper bound for the quantity inside the supremum. By denoting the expectations of $\hat{\mu}_2(\beta)$ and $\hat{\mu}_4(\beta)$, with $\mu_2(\beta)$ and $\mu_4(\beta)$ respectively, we write

$$\left\| \hat{\mu}_2(\beta) \zeta_r(\widehat{\mathbf{\Sigma}}_S) + \hat{\mu}_4(\beta) \zeta_r(\widehat{\mathbf{\Sigma}}_S) \beta (\zeta_r(\widehat{\mathbf{\Sigma}}_S) \beta)^T - \mathbb{E}[\phi^{(2)}(\langle x, \beta \rangle)] \mathbf{\Sigma} - \mathbb{E}[\phi^{(4)}(\langle x, \beta \rangle)] \mathbf{\Sigma} \beta (\mathbf{\Sigma} \beta)^T \right\|_2$$
$$\leq \left\| \hat{\mu}_2(\beta) \zeta_r(\widehat{\mathbf{\Sigma}}_S) - \mu_2(\beta) \mathbf{\Sigma} \right\|_2 + \left\| \hat{\mu}_4(\beta) \zeta_r(\widehat{\mathbf{\Sigma}}_S) \beta (\zeta_r(\widehat{\mathbf{\Sigma}}_S) \beta)^T - \mu_4(\beta) \mathbf{\Sigma} \beta (\mathbf{\Sigma} \beta)^T \right\|_2.$$

For the first term on the right hand side, we have

$$\left\| \hat{\mu}_2(\beta) \zeta_r(\widehat{\mathbf{\Sigma}}_S) - \mu_2(\beta) \mathbf{\Sigma} \right\|_2 \leq |\hat{\mu}_2(\beta)| \left\| \zeta_r(\widehat{\mathbf{\Sigma}}_S) - \mathbf{\Sigma} \right\|_2 + \|\mathbf{\Sigma}\|_2 |\hat{\mu}_2(\beta) - \mu_2(\beta)|,$$
$$\leq B_2 \left\| \zeta_r(\widehat{\mathbf{\Sigma}}_S) - \mathbf{\Sigma} \right\|_2 + K |\hat{\mu}_2(\beta) - \mu_2(\beta)|.$$

By the Lemmas B.1 and B.2, for some constants $c_1, c_2, c_3$, we have with probability $1 - c_2 e^{-c_3 p} - 1/p^2$,

$$\sup_{\{\beta \in B_p(R)\}} \left\| \hat{\mu}_2(\beta) \zeta_r(\widehat{\mathbf{\Sigma}}_S) - \mu_2(\beta) \mathbf{\Sigma} \right\|_2 \leq 2 B_2 K \sqrt{C} \sqrt{\frac{\log(p)}{|S|}} + c_1 K \sqrt{\frac{p \log(n)}{n}}$$
$$= \mathcal{O} \left( \sqrt{\frac{p}{\min \{p/\log(p)|S|, n/\log(n)\}}} \right).$$

For the second term, we have

$$\left\| \hat{\mu}_4(\beta) \zeta_r(\widehat{\mathbf{\Sigma}}_S) \beta (\zeta_r(\widehat{\mathbf{\Sigma}}_S) \beta)^T - \mu_4(\beta) \mathbf{\Sigma} \beta (\mathbf{\Sigma} \beta)^T \right\|_2$$
$$\leq |\hat{\mu}_4(\beta)| \left\| \zeta_r(\widehat{\mathbf{\Sigma}}_S) \beta \beta^T \zeta_r(\widehat{\mathbf{\Sigma}}_S) - \mathbf{\Sigma} \beta \beta^T \mathbf{\Sigma} \right\|_2 + |\hat{\mu}_4(\beta) - \mu_4(\beta)| \left\| \mathbf{\Sigma} \beta \beta^T \mathbf{\Sigma} \right\|_2,$$
$$\leq B_4 R^2 \left\{ \|\zeta_r(\widehat{\mathbf{\Sigma}}_S)\|_2 + \|\mathbf{\Sigma}\|_2 \right\} \left\| \zeta_r(\widehat{\mathbf{\Sigma}}_S) - \mathbf{\Sigma} \right\|_2 + R^2 \|\mathbf{\Sigma}\|_2^2 |\hat{\mu}_4(\beta) - \mu_4(\beta)|,$$
$$\leq B_4 R^2 \left\{ \|\zeta_r(\widehat{\mathbf{\Sigma}}_S)\|_2 + K \right\} \left\| \zeta_r(\widehat{\mathbf{\Sigma}}_S) - \mathbf{\Sigma} \right\|_2 + R^2 K^2 |\hat{\mu}_4(\beta) - \mu_4(\beta)|.$$

Again, by the Lemmas B.1, B.2 and C.2, for some constants $c_1, c_2, c_3$, we have with probability $1 - c_2 e^{-c_3 p} - 1/p^2$, we write

$$B_4 R^2 \left\{ \|\zeta_r(\widehat{\mathbf{\Sigma}}_S)\|_2 + K \right\} \left\| \zeta_r(\widehat{\mathbf{\Sigma}}_S) - \mathbf{\Sigma} \right\|_2 \leq 2 K \sqrt{C} B_4 R^2 \left\{ 2K + 3K \sqrt{C} \sqrt{\frac{\log(p)}{|S|}} \right\} \sqrt{\frac{\log(p)}{|S|}},$$
$$\leq 4 K^2 \sqrt{C} B_4 R^2 \sqrt{\frac{\log(p)}{|S|}} + 6 K^2 C B_4 R^2 \frac{\log(p)}{|S|},$$
$$= \mathcal{O} \left( \sqrt{\frac{\log(p)}{|S|}} \right),$$

for sufficiently large $|S|$.

Further, by Lemma C.1, for constants $c_1, c_2, c_3$, we have with probability $1 - c_2 e^{-c_3 p}$,

$$\sup_{\{\beta \in B_p(R)\}} |\hat{\mu}_4(\beta) - \mu_4(\beta)| \le c_1 \sqrt{\frac{p \log(n)}{n}} = \mathcal{O}\left(\sqrt{\frac{p \log(n)}{n}}\right).$$

Combining the above results, for sufficiently large $p, |S|$ and constants $c_1, c_2$, we have with probability at least $1 - c_1/p^2$,

$$\sup_{\{\beta \in B_p(R)\}} \left\| \hat{\mu}_4(\beta) \zeta_r(\widehat{\boldsymbol{\Sigma}}_S) \beta (\zeta_r(\widehat{\boldsymbol{\Sigma}}_S)\beta)^T - \mu_4(\beta) \boldsymbol{\Sigma} \beta (\boldsymbol{\Sigma}\beta)^T \right\|_2$$

$$\le 4K^2 \sqrt{C} \max\{B_2, B_4\} R^2 \sqrt{\frac{\log(p)}{|S|}} + 6K^2 C B_4 R^2 \frac{\log(p)}{|S|} + c_1 R^2 K^2 \sqrt{\frac{p \log(n)}{n}}$$

$$= \mathcal{O}\left(\sqrt{\frac{p}{\min\{|S|p/\log(p), n/\log(n)\}}}\right).$$

Hence, for some constants $c, C$, with probability $1 - c/p^2$, we have

$$\left\| [\mathbf{Q}^t]^{-1} - \mathfrak{E}(\hat{\beta}^t) \right\|_2 \le C \sqrt{\frac{p}{\min\{|S|p/\log(p), n/\log(n)\}}},$$

where the constants depend on $K, B = \max\{B_2, B_4\}$ and the radius $R$. $\qquad\square$

**Lemma D.2.** *The bias term can be upper bounded by*

$$\left\| \mathbb{E}[xx^T \phi^{(2)}(\langle x, \hat{\beta}^t \rangle)] - \mathfrak{E}(\hat{\beta}^t) \right\|_2 \le d_{\mathcal{H}_3}(x, z) + \|\boldsymbol{\Sigma}\|_2 \, d_{\mathcal{H}_1}(x, z) + \|\boldsymbol{\Sigma}\|_2^2 R^2 \, d_{\mathcal{H}_2}(x, z).$$

*Proof of Lemma D.2.* For a random variable $z \sim \mathsf{N}_p(0, \boldsymbol{\Sigma})$, by the triangle inequality, we write

$$\left\| \mathbb{E}[xx^T \phi^{(2)}(\langle x, \hat{\beta}^t \rangle)] - \mathfrak{E}(\hat{\beta}^t) \right\|_2$$

$$\le \left\| \mathbb{E}[xx^T \phi^{(2)}(\langle x, \hat{\beta}^t \rangle)] - \mathbb{E}[zz^T \phi^{(2)}(\langle z, \hat{\beta}^t \rangle)] \right\|_2 + \left\| \mathbb{E}[zz^T \phi^{(2)}(\langle z, \hat{\beta}^t \rangle)] - \mathfrak{E}(\hat{\beta}^t) \right\|_2$$

For the first term on the right hand side, we have

$$\left\| \mathbb{E}[xx^T \phi^{(2)}(\langle x, \hat{\beta}^t \rangle)] - \mathbb{E}[zz^T \phi^{(2)}(\langle z, \hat{\beta}^t \rangle)] \right\|_2$$

$$\le \sup_{\beta \in B_p(R)} \sup_{\|v\|_2 = 1} \left| \mathbb{E}\left[ \langle v, x \rangle^2 \phi^{(2)}(\langle x, \beta \rangle) \right] - \mathbb{E}\left[ \langle v, z \rangle^2 \phi^{(2)}(\langle z, \beta \rangle) \right] \right|,$$

$$\le d_{\mathcal{H}_3}(x, z).$$

For the second term, we write

$$\left\| [\mathbb{E}[zz^T \phi^{(2)}(\langle z, \hat{\beta}^t \rangle)] - \mathfrak{E}(\hat{\beta}^t) \right\|_2$$

$$\leq \sup_{\{\beta \in B_p(R)\}} \left\| \mathbb{E}[zz^T \phi^{(2)}(\langle z, \beta \rangle)] - \mathbb{E}[\phi^{(2)}(\langle x, \beta \rangle)] \mathbf{\Sigma} + \mathbb{E}\left[ \phi^{(4)}(\langle x, \beta \rangle) \right] \mathbf{\Sigma} \beta \beta^T \mathbf{\Sigma} \right\|_2,$$

$$\leq \sup_{\{\beta \in B_p(R)\}} \left\| \mathbb{E}[\phi^{(2)}(\langle z, \beta \rangle)] \mathbf{\Sigma} + \mathbb{E}\left[ \phi^{(4)}(\langle z, \beta \rangle) \right] \mathbf{\Sigma} \beta \beta^T \mathbf{\Sigma} \right.$$

$$\left. - \mathbb{E}[\phi^{(2)}(\langle x, \beta \rangle)] \mathbf{\Sigma} - \mathbb{E}\left[ \phi^{(4)}(\langle x, \beta \rangle) \right] \mathbf{\Sigma} \beta \beta^T \mathbf{\Sigma} \right\|_2,$$

$$\leq \sup_{\{\beta \in B_p(R)\}} \left\| \mathbb{E}[\phi^{(2)}(\langle z, \beta \rangle)] \mathbf{\Sigma} - \mathbb{E}[\phi^{(2)}(\langle x, \beta \rangle)] \mathbf{\Sigma} \right\|_2,$$

$$+ \sup_{\{\beta \in B_p(R)\}} \left\| \mathbb{E}\left[ \phi^{(4)}(\langle z, \beta \rangle) \right] \mathbf{\Sigma} \beta \beta^T \mathbf{\Sigma} - \mathbb{E}\left[ \phi^{(4)}(\langle x, \beta \rangle) \right] \mathbf{\Sigma} \beta \beta^T \mathbf{\Sigma} \right\|_2,$$

$$\leq \|\mathbf{\Sigma}\|_2 \sup_{\{\beta \in B_p(R)\}} \left| \mathbb{E}[\phi^{(2)}(\langle z, \beta \rangle)] - \mathbb{E}[\phi^{(2)}(\langle x, \beta \rangle)] \right|$$

$$+ \|\mathbf{\Sigma}\|_2^2 R^2 \sup_{\{\beta \in B_p(R)\}} \left| \mathbb{E}[\phi^{(4)}(\langle z, \beta \rangle)] - \mathbb{E}[\phi^{(4)}(\langle x, \beta \rangle)] \right|,$$

$$\leq \|\mathbf{\Sigma}\|_2 d_{\mathcal{H}_1}(x, z) + \|\mathbf{\Sigma}\|_2^2 R^2 d_{\mathcal{H}_2}(x, z).$$

Hence, we conclude that

$$\left\| \mathbb{E}[xx^T \phi^{(2)}(\langle x, \hat{\beta}^t \rangle)] - \mathfrak{E}(\hat{\beta}^t) \right\|_2 \leq d_{\mathcal{H}_3}(x, z) + \|\mathbf{\Sigma}\|_2 \, d_{\mathcal{H}_1}(x, z) + \|\mathbf{\Sigma}\|_2^2 R^2 \, d_{\mathcal{H}_2}(x, z).$$

$$\square$$

**Lemma D.3.** *There exist constants $c_1, c_2, c_3$ depending on $K$, $B$, $L$ and $R$ such that, with probability at least $1 - c_2 e^{-c_3 p}$*

$$\left\| \frac{1}{n} \sum_{i=1}^n x_i x_i^T \int_0^1 \phi^{(2)}(\langle x_i, \beta_* + \xi(\hat{\beta}^t - \beta_*) \rangle) d\xi - \mathbb{E}\left[ xx^T \int_0^1 \phi^{(2)}(\langle x, \beta_* + \xi(\hat{\beta}^t - \beta_*) \rangle) d\xi \right] \right\|_2$$

$$\leq c_1 \sqrt{\frac{p}{n} \log(n)}.$$

*Proof.* By the Fubini's theorem, we have

$$\left\| \frac{1}{n} \sum_{i=1}^n x_i x_i^T \int_0^1 \phi^{(2)}(\langle x_i, \beta_* + \xi(\hat{\beta}^t - \beta_*) \rangle) d\xi - \mathbb{E}\left[ xx^T \int_0^1 \phi^{(2)}(\langle x, \beta_* + \xi(\hat{\beta}^t - \beta_*) \rangle) d\xi \right] \right\|_2,$$

$$= \left\| \int_0^1 \left\{ \frac{1}{n} \sum_{i=1}^n x_i x_i^T \phi^{(2)}(\langle x_i, \beta_* + \xi(\hat{\beta}^t - \beta_*) \rangle) - \mathbb{E}\left[ xx^T \phi^{(2)}(\langle x, \beta_* + \xi(\hat{\beta}^t - \beta_*) \rangle) \right] \right\} d\xi \right\|_2,$$

$$\leq \int_0^1 \left\| \left\{ \frac{1}{n} \sum_{i=1}^n x_i x_i^T \phi^{(2)}(\langle x_i, \beta_* + \xi(\hat{\beta}^t - \beta_*) \rangle) - \mathbb{E}\left[ xx^T \phi^{(2)}(\langle x, \beta_* + \xi(\hat{\beta}^t - \beta_*) \rangle) \right] \right\} \right\|_2 d\xi,$$

$$\leq \sup_{\beta \in B_p(R)} \left\| \frac{1}{n} \sum_{i=1}^n x_i x_i^T \phi^{(2)}(\langle x_i, \beta \rangle) - \mathbb{E}\left[ xx^T \phi^{(2)}(\langle x, \beta \rangle) \right] \right\|_2.$$

Using the definition of *operator norm*, the right hand side is equal to

$$\sup_{\beta \in B_p(R)} \left\| \frac{1}{n}\sum_{i=1}^{n} x_i x_i^T \phi^{(2)}(\langle x_i, \beta\rangle) - \mathbb{E}\left[ xx^T \phi^{(2)}(\langle x, \beta\rangle)\right]\right\|_2$$

$$= \sup_{\beta \in B_p(R)} \sup_{v \in S^{p-1}} \left| \frac{1}{n}\sum_{i=1}^{n} \phi^{(2)}(\langle x_i, \beta\rangle)\langle x_i, v\rangle^2 - \mathbb{E}\left[ \phi^{(2)}(\langle x, \beta\rangle)\langle x, v\rangle^2\right]\right|,$$

where $S^{p-1}$ denotes the $p$-dimensional unit sphere.

For $\Delta = 0.25$, let $T_\Delta$ be an $\Delta$-net over $S^{p-1}$. Using Lemma E.1, we obtain

$$\mathbb{P}\left( \sup_{\beta \in B_p(R)} \sup_{v \in S^{p-1}} \left| \frac{1}{n}\sum_{i=1}^{n} \phi^{(2)}(\langle x_i, \beta\rangle)\langle x_i, v\rangle^2 - \mathbb{E}\left[ \phi^{(2)}(\langle x, \beta\rangle)\langle x, v\rangle^2\right]\right| > \epsilon \right),$$

$$\leq \mathbb{P}\left( \sup_{\beta \in B_p(R)} \sup_{v \in T_\Delta} \left| \frac{1}{n}\sum_{i=1}^{n} \phi^{(2)}(\langle x_i, \beta\rangle)\langle x_i, v\rangle^2 - \mathbb{E}\left[ \phi^{(2)}(\langle x, \beta\rangle)\langle x, v\rangle^2\right]\right| > \epsilon/2 \right),$$

$$\leq |T_\Delta| \mathbb{P}\left( \sup_{\beta \in B_p(R)} \left| \frac{1}{n}\sum_{i=1}^{n} \phi^{(2)}(\langle x_i, \beta\rangle)\langle x_i, v\rangle^2 - \mathbb{E}\left[ \phi^{(2)}(\langle x, \beta\rangle)\langle x, v\rangle^2\right]\right| > \epsilon/2 \right),$$

$$= 9^p \mathbb{P}\left( \sup_{\beta \in B_p(R)} \left| \frac{1}{n}\sum_{i=1}^{n} \phi^{(2)}(\langle x_i, \beta\rangle)\langle x_i, v\rangle^2 - \mathbb{E}\left[ \phi^{(2)}(\langle x, \beta\rangle)\langle x, v\rangle^2\right]\right| > \epsilon/2 \right).$$

By applying Lemma C.2 to the last line above, there exists absolute constants $c_1', c_2', c_3'$ depending on $L, B, R, K$ such that, we have

$$\mathbb{P}\left( \sup_{\beta \in B_p(R)} \left| \frac{1}{n}\sum_{i=1}^{n} \phi^{(2)}(\langle x_i, \beta\rangle)\langle x_i, v\rangle^2 - \mathbb{E}[\phi^{(2)}(\langle x, \beta\rangle)\langle x, v\rangle^2]\right| > c_1'\sqrt{\frac{p}{n}\log(n)} \right) \leq c_2' e^{-c_3' p}.$$

$c_3'$ is of order $\mathcal{O}(p \log\log(n))$. Therefore, by choosing $n$ large enough, we obtain that there exists constants $c_1, c_2, c_3$ such that with probability at least $1 - c_2 e^{-c_3 p}$

$$\sup_{\beta \in B} \left\| \frac{1}{n}\sum_{i=1}^{n} x_i x_i^T \phi^{(2)}(\langle x_i, \beta\rangle) - \mathbb{E}\left[ xx^T \phi^{(2)}(\langle x, \beta\rangle)\right]\right\|_2 \leq c_1 \sqrt{\frac{p}{n}\log(n)}$$

$\square$

**Lemma D.4.** *There exists a constant $C$ depending on $K$ and $L$ such that,*

$$\left\| \mathbb{E}[xx^T \phi^{(2)}(\langle x, \hat{\beta}^t\rangle)] - \mathbb{E}\left[ xx^T \int_0^1 \phi^{(2)}(\langle x, \beta_* + \xi(\hat{\beta}^t - \beta_*)\rangle)d\xi\right]\right\|_2 \leq C\|\hat{\beta}^t - \beta_*\|_2.$$

*Proof.* By the Fubini's theorem, we write

$$\left\| \mathbb{E}[xx^T \phi^{(2)}(\langle x, \hat{\beta}^t\rangle)] - \mathbb{E}\left[ xx^T \int_0^1 \phi^{(2)}(\langle x, \beta_* + \xi(\hat{\beta}^t - \beta_*)\rangle)d\xi\right]\right\|_2,$$

$$= \left\| \int_0^1 \mathbb{E}\left[ xx^T \left\{ \phi^{(2)}(\langle x, \hat{\beta}^t\rangle) - \phi^{(2)}(\langle x, \beta_* + \xi(\hat{\beta}^t - \beta_*)\rangle)\right\}\right]d\xi\right\|_2,$$

Moving the integration out, right hand side of above equation is smaller than

$$\int_0^1 \left\| \mathbb{E}\left[ xx^T \left\{ \phi^{(2)}(\langle x, \hat{\beta}^t \rangle) - \phi^{(2)}(\langle x, \beta_* + \xi(\hat{\beta}^t - \beta_*)\rangle) \right\} \right] \right\|_2 d\xi,$$

$$\leq \int_0^1 \left\| \mathbb{E}\left[ xx^T L |\langle x, (1-\xi)(\hat{\beta}^t - \beta_*)\rangle| \right] \right\|_2 d\xi,$$

$$\leq \mathbb{E}\left[ \|x\|_2^3 \|\hat{\beta}^t - \beta_*\|_2 \right] L \int_0^1 (1-\xi) d\xi,$$

$$= \frac{LK^{3/2}}{2} \|\hat{\beta}^t - \beta_*\|_2.$$

$\square$

By combining above results, we obtain

$$\left\| [\mathbf{Q}^t]^{-1} - \int_0^1 \mathbf{\nabla}_\beta^2 \ell(\beta_* + \xi(\hat{\beta}^t - \beta_*)) d\xi \right\|_2$$

$$\leq \mathfrak{D}(x, z) + c_1 \sqrt{\frac{p}{\min\{|S|p/\log(p), n/\log(n)\}}} + c_2 \|\hat{\beta}^t - \beta_*\|_2,$$

where

$$\mathfrak{D}(x, z) = d_{\mathcal{H}_3}(x, z) + \|\mathbf{\Sigma}\|_2\, d_{\mathcal{H}_1}(x, z) + \|\mathbf{\Sigma}\|_2^2 R^2\, d_{\mathcal{H}_2}(x, z).$$

In the following, we will derive an upper bound for $\|\mathbf{Q}^t\|_2$ where,

$$\mathbf{Q}^t = \frac{1}{\hat{\mu}_2(\hat{\beta}^t)} \left[ \zeta_r(\widehat{\mathbf{\Sigma}}_S)^{-1} - \frac{\hat{\beta}^t[\hat{\beta}^t]^T}{\hat{\mu}_2(\hat{\beta}^t)/\hat{\mu}_4(\hat{\beta}^t) + \langle \zeta_r(\widehat{\mathbf{\Sigma}}_S)\hat{\beta}^t, \hat{\beta}^t \rangle} \right].$$

We define

$$c_L = \inf_{\beta \in B_p(L)} \mu_2(\beta).$$

Thus, for any iterate $\hat{\beta}^t$ of Newton-Stein algorithm

$$\mu_2(\hat{\beta}^t) \geq c_R.$$

By Lemma C.1, for some constants $c_1, c_2, c_3$, with probability $1 - c_2 e^{-c_3 p}$,

$$\hat{\mu}_2(\hat{\beta}^t) \geq \mu_2(\hat{\beta}^t) - c_1 \sqrt{\frac{p \log(n)}{n}},$$

$$\geq c_R - c_1 \sqrt{\frac{p \log(n)}{n}}.$$

Also, by the assumption given in the theorem, on the set $\mathcal{E}$ we have almost surely,

$$\inf_{t \geq 0} \left| \mu_2(\hat{\beta}^t) + \mu_4(\hat{\beta}^t) \langle \mathbf{\Sigma}\hat{\beta}^t, \hat{\beta}^t \rangle \right| > \xi,$$

for some $\xi > 0$. With probability at least $1 - c_2 e^{-c_3 p}$,

$$\left| \hat{\mu}_2(\hat{\beta}^t) + \hat{\mu}_4(\hat{\beta}^t) \langle \zeta_r(\widehat{\mathbf{\Sigma}}_S)\hat{\beta}^t, \hat{\beta}^t \rangle \right| \geq \left| \mu_2(\hat{\beta}^t) + \mu_4(\hat{\beta}^t) \langle \mathbf{\Sigma}\hat{\beta}^t, \hat{\beta}^t \rangle \right| - \left\{ \left| \hat{\mu}_2(\hat{\beta}^t) - \mu_2(\hat{\beta}^t) \right| \right.$$

$$+ \left| \mu_4(\hat{\beta}^t) \langle \mathbf{\Sigma}\hat{\beta}^t, \hat{\beta}^t \rangle - \hat{\mu}_4(\hat{\beta}^t) \langle \mathbf{\Sigma}\hat{\beta}^t, \hat{\beta}^t \rangle \right|$$

$$\left. + \left| \hat{\mu}_4(\hat{\beta}^t) \langle \mathbf{\Sigma}\hat{\beta}^t, \hat{\beta}^t \rangle - \hat{\mu}_4(\hat{\beta}^t) \langle \zeta_r(\widehat{\mathbf{\Sigma}}_S)\hat{\beta}^t, \hat{\beta}^t \rangle \right| \right\}.$$

By the Lemmas B.1 and C.1, we have

$$\left|\hat{\mu}_2(\hat{\beta}^t) + \hat{\mu}_4(\hat{\beta}^t)\langle\zeta_r(\widehat{\boldsymbol{\Sigma}}_S)\hat{\beta}^t, \hat{\beta}^t\rangle\right| \geq \left|\mu_2(\hat{\beta}^t) + \mu_4(\hat{\beta}^t)\langle\boldsymbol{\Sigma}\hat{\beta}^t, \hat{\beta}^t\rangle\right| - \left(c_1 + \|\hat{\beta}^t\|_2^2\|\boldsymbol{\Sigma}\|_2\right)\sqrt{\frac{p\log(n)}{n}}$$

$$- B_4\|\hat{\beta}^t\|_2^2\left\|\zeta_r(\widehat{\boldsymbol{\Sigma}}_S) - \boldsymbol{\Sigma}\right\|_2,$$

$$\geq \left|\mu_2(\hat{\beta}^t) + \mu_4(\hat{\beta}^t)\langle\boldsymbol{\Sigma}\hat{\beta}^t, \hat{\beta}^t\rangle\right| - C\sqrt{\frac{p}{\min\{n/\log(n), p/\log(p)|S|\}}},$$

$$\geq \xi - C\sqrt{\frac{p}{\min\{n/\log(n), p/\log(p)|S|\}}},$$

where $C = \max\{cB_4R^2, c_1 + R^2\|\boldsymbol{\Sigma}\|_2\}$.

Therefore, for some constants $c_1, c_2, c_3, c_4$, with probability $1 - c_2 e^{-c_3 p} - c/p^2$, we have

$$\|\mathbf{Q}^t\|_2 \leq \frac{1}{\hat{\mu}_2(\hat{\beta}^t)}\left[\left\|\zeta_r(\widehat{\boldsymbol{\Sigma}}_S)^{-1}\right\|_2 + \frac{|\hat{\mu}_4(\hat{\beta}^t)|\|\hat{\beta}^t\|_2^2}{\left|\hat{\mu}_2(\hat{\beta}^t) + \hat{\mu}_4(\hat{\beta}^t)\langle\zeta_r(\widehat{\boldsymbol{\Sigma}}_S)\hat{\beta}^t, \hat{\beta}^t\rangle\right|}\right],$$

$$\leq \frac{1}{c_R - c_1\sqrt{\frac{p\log(n)}{n}}}\left[\frac{1}{\hat{\sigma}^2} + \frac{B_4R^2}{\xi - C\sqrt{\frac{p}{\min\{n/\log(n), p/\log(p)|S|\}}}}\right],$$

$$\leq \frac{1}{c_R - c_1\sqrt{\frac{p\log(n)}{n}}}\left[\frac{1}{\sigma^2 - c_4\sqrt{\frac{\log(p)}{|S|}}} + \frac{B_4R^2}{\xi - C\sqrt{\frac{p}{\min\{n/\log(n), p/\log(p)|S|\}}}}\right],$$

For $n$ and $|S|$ sufficiently large so that we have the following inequalities,

$$c_4\sqrt{\frac{\log(p)}{|S|}} \leq \frac{\sigma^2}{2},$$

$$c_1\sqrt{\frac{p\log(n)}{n}} \leq \frac{c_R}{2},$$

$$C\sqrt{\frac{p}{\min\{n/\log(n), |S|\}}} \leq \frac{\xi}{2},$$

we obtain

$$\|\mathbf{Q}^t\|_2 \leq \frac{2}{c_R}\left[\frac{2}{\sigma^2} + \frac{2B_4R^2}{\xi}\right] := \kappa.$$

Finally, we take into account the conditioning on the event $\mathcal{E}$ and conclude the proof.

*Proof of Corollary 4.2.* The statement of the Theorem 4.1 holds on the probability space with a probability lower bounded by $\mathbb{P}(\mathcal{E}) - c/p^2$ for some constant $c$. Let $\mathcal{Q}$ denote this set, on which the statement of the lemma holds. Note that $\mathcal{Q} \subset \mathcal{E}$. We have

$$\mathbb{P}(\mathcal{Q}) \geq \mathbb{P}(\mathcal{E}) - c'/p^2.$$

This suggests that the difference between $\mathcal{Q}$ and $\mathcal{E}$ is small. By taking expectations on both sides over the set $\mathcal{Q}$, we obtain,

$$\mathbb{E}\left[\|\hat{\beta}^{t+1} - \beta_*\|_2; \mathcal{Q}\right] \leq \kappa\left\{\mathfrak{D}(x, z) + c_1\sqrt{\frac{p}{\min\{p/\log(p)|S|, n/\log(n)\}}}\right\}\mathbb{E}\left[\|\hat{\beta}^t - \beta_*\|_2\right]$$

$$+ \kappa c_2\mathbb{E}\left[\|\hat{\beta}^t - \beta_*\|_2^2\right]$$

where we used

$$\mathbb{E}\left[\|\hat{\beta}^t - \beta_*\|_2^l; \mathcal{Q}\right] \le \mathbb{E}\left[\|\hat{\beta}^t - \beta_*\|_2^l\right], \quad l = 1, 2.$$

Similarly for the iterate $\hat{\beta}^{t+1}$, we write

$$
\begin{aligned}
\mathbb{E}\left[\|\hat{\beta}^{t+1} - \beta_*\|_2\right] &= \mathbb{E}\left[\|\hat{\beta}^{t+1} - \beta_*\|_2; \mathcal{Q}\right] + \mathbb{E}\left[\|\hat{\beta}^{t+1} - \beta_*\|_2; \mathcal{Q}^C\right], \\
&\le \mathbb{E}\left[\|\hat{\beta}^{t+1} - \beta_*\|_2; \mathcal{Q}\right] + 2R\mathbb{P}(\mathcal{Q}^C), \\
&\le \mathbb{E}\left[\|\hat{\beta}^{t+1} - \beta_*\|_2; \mathcal{Q}\right] + 2R\left(\mathbb{P}(\mathcal{E}^C) + \frac{c}{p^2}\right), \\
&\le \mathbb{E}\left[\|\hat{\beta}^{t+1} - \beta_*\|_2; \mathcal{Q}\right] + \frac{\epsilon}{10}, \\
&\le \mathbb{E}\left[\|\hat{\beta}^{t+1} - \beta_*\|_2; \mathcal{Q}\right] + \frac{\mathbb{E}\left[\|\hat{\beta}^t - \beta_*\|_2\right]}{10}.
\end{aligned}
$$

Combining these two inequalities, we obtain

$$\mathbb{E}\left[\|\hat{\beta}^{t+1} - \beta_*\|_2\right] \le \left\{0.1 + \kappa\mathfrak{D}(x,z) + c_1\kappa\sqrt{\frac{p}{\min\left\{p/\log(p)|S|, n/\log(n)\right\}}}\right\}\mathbb{E}\left[\|\hat{\beta}^t - \beta_*\|_2\right]$$
$$+ c_2\kappa\mathbb{E}\left[\|\hat{\beta}^t - \beta_*\|_2^2\right].$$

Hence the proof follows. $\square$

*Proof of Theorem 4.3.* For a sequence satisfying the following inequality,

$$\|\hat{\beta}^{t+1} - \beta_*\|_2 \le \left(\tau_1 + \tau_2\|\hat{\beta}^t - \beta_*\|_2\right)\|\hat{\beta}^t - \beta_*\|_2,$$

we observe that

$$\tau_1 + \tau_2\|\hat{\beta}^0 - \beta_*\|_2 < 1 \tag{D.2}$$

is a sufficient condition for convergence to 0. Let $\xi \in (\epsilon, 1)$ and $t_\xi$ be the last iteration that $\|\hat{\beta}^t - \beta_*\|_2 > \delta$. Then, for $t > t_\xi$

$$
\begin{aligned}
\|\hat{\beta}^{t+1} - \beta_*\|_2 &\le \left(\tau_1 + \tau_2\|\hat{\beta}^t - \beta_*\|_2\right)\|\hat{\beta}^t - \beta_*\|_2, \\
&\le (\tau_1 + \tau_2\xi)\|\hat{\beta}^t - \beta_*\|_2.
\end{aligned}
$$

This convergence behavior describes a linear rate and requires at most

$$\frac{\log(\epsilon/\xi)}{\log(\tau_1 + \tau_2\xi)}$$

iterations to reach a tolerance of $\epsilon$. For $t \le t_\xi$, we have

$$
\begin{aligned}
\|\hat{\beta}^{t+1} - \beta_*\|_2 &\le \left(\tau_1 + \tau_2\|\hat{\beta}^t - \beta_*\|_2\right)\|\hat{\beta}^t - \beta_*\|_2, \\
&\le (\tau_1/\xi + \tau_2)\|\hat{\beta}^t - \beta_*\|_2^2.
\end{aligned}
$$

This describes a quadratic rate and the number of iterations to reach a tolerance of $\xi$ can be upper bounded by

$$\log_2\left(\frac{\log\left(\delta\left(\tau_1/\xi + \tau_2\right)\right)}{\log\left(\left(\tau_1/\xi + \tau_2\right)\right)\|\hat{\beta}^0 - \beta_*\|_2}\right).$$

Therefore, the overall number of iterations to reach a tolerance of $\epsilon$ is upper bounded by

$$\mathcal{J}(\xi) = \log_2\left(\frac{\log\left(\delta\left(\tau_1/\xi + \tau_2\right)\right)}{\log\left(\left(\tau_1/\xi + \tau_2\right)\right)\|\hat{\beta}^0 - \beta_*\|_2}\right) + \frac{\log(\epsilon/\xi)}{\log(\tau_1 + \tau_2\xi)}$$

which is a function of $\xi$. Therefore, we take the minimum over the feasible set. $\square$

# E    Useful lemmas

**Lemma E.1** ([Ver10]). *Let $X$ be a symmetric $p \times p$ matrix, and let $T_\epsilon$ be an $\epsilon$-net over $S^{p-1}$. Then,*

$$\|X\|_2 \le \frac{1}{1 - 2\epsilon} \sup_{v \in T_\epsilon} |\langle Xv, v \rangle|.$$

**Lemma E.2.** *Let $B_p(R) \subset \mathbb{R}^p$ be the ball of radius $R$ centered at the origin and $T_\epsilon$ be an $\epsilon$-net over $B_p(R)$. Then,*

$$|T_\epsilon| \le \left( \frac{R\sqrt{p}}{\epsilon} \right)^p.$$

*Proof of Lemma E.2.* The set $B_p(R)$ can be contained in a $p$-dimensional cube of size $2R$. Consider a grid over this cube with mesh width $2\epsilon/\sqrt{p}$. Then $B_p(R)$ can be covered with at most $(2R/(2\epsilon/\sqrt{p}))^p$ many cubes of edge length $2\epsilon/\sqrt{p}$. If ones takes the projection of the centers of such cubes onto $B_p(R)$ and considers the circumscribed balls of radius $\epsilon$, we may conclude that $B_p(R)$ can be covered with at most

$$\left( \frac{2R}{2\epsilon/\sqrt{p}} \right)^p$$

many balls of radius $\epsilon$. $\square$

**Lemma E.3.** *For $a, b > 0$, and $\epsilon$ satisfying*

$$\epsilon = \left\{ \frac{a}{2} \log \left( \frac{2b^2}{a} \right) \right\}^{1/2} \quad \text{and} \quad \frac{2}{a} b^2 > e,$$

*we have $\epsilon^2 \ge a \log(b/\epsilon)$.*

*Proof of Lemma E.3.* Since $a, b > 0$ and $x \to e^x$ is a monotone increasing function, the above inequality condition is equivalent to

$$\frac{2\epsilon^2}{a} e^{\frac{2\epsilon^2}{a}} \ge \frac{2b^2}{a}.$$

Now, we define the function $f(w) = we^w$ for $w > 0$. $f$ is continuous and invertible on $[0, \infty)$. Note that $f^{-1}$ is also a continuous and increasing function for $w > 0$. Therefore, we have

$$\epsilon^2 \ge \frac{a}{2} f^{-1} \left( \frac{2b^2}{a} \right)$$

Observe that the smallest possible value for $\epsilon$ would be simply the square root of $a f^{-1} \left( 2b^2/a \right)/2$. For simplicity, we will obtain a more interpretable expression for $\epsilon$. By the definition of $f^{-1}$, we have

$$\log(f^{-1}(y)) + f^{-1}(y) = \log(y).$$

Since the condition on $a$ and $b$ enforces $f^{-1}(y)$ to be larger than 1, we obtain the simple inequality that

$$f^{-1}(y) \le \log(y).$$

Using the above inequality, if $\epsilon$ satisfies

$$\epsilon^2 = \frac{a}{2} \log \left( \frac{2b^2}{a} \right) \ge \frac{a}{2} f^{-1} \left( \frac{2b^2}{a} \right),$$

we obtain the desired inequality. $\square$