[Reviews · NeurIPS 2015]

Submitted by Assigned_Reviewer_1

This paper considers the optimization problem for generalized linear models. The main idea is to incorporate some curvature information.
Summary: This paper introduces a new idea on incorporating second order info when calculating the MLE for GLMs. The ideas seem to be interesting.

Submitted by Assigned_Reviewer_2

the paper proposes a quasi-newton method to compute the MLE of a Generalized Linear Model. The insight behind the paper is this:

When the covariates are Gaussian, and under mild conditions on the link function, the Hessian of the GLM log likelihood decomposes into a weighted sum of a constant full rank matrix that can be computed ahead of time, and a rank one matrix that changes at each iteration. The Newton step can therefore be quickly computed using the Sherman-Morrison formula. This is not an approximation, but rather a consequence of the same trick used to prove bounds for the James-Stein estimator.

In reality, covariates are not Gaussian, but it is claimed that the convergence rate degrades gently for other distributions in a way that depends on the distance between the distribution and its moment-matched Gaussian counterpart.

Experiments: The authors should compare against a Natural Gradient Descent as well.

Otherwise, the experiments are convincing enough.

The method seems to outperform other second order methods on a few datasets.

The theory section, which derives the convergence rate for non-Gaussian covariates doesn't provide me much insight because proofs are relegated to supplemental material. in the absence of proofs, the section would be more instructive to me if it went like this: theorem with convergence rate for Gaussian covariates, with a proof outline. Then identify where the proof breaks down for non-Gaussian covariates and how it can be patched up with a Gaussian surrogate and the TV-like distance. To save space, i would be ok relegating the more general but weaker subgaussian case to supplemental material and focus on the bounded case, which handles probably all the cases any way (the give away was the unsued psi_2 norm in the notation section, which forshadows a standard maneuver in the transition from bounded to subgaussian)
Summary: exploits a very clever trick to compute very cheap quasi-newton method. experiments are convincing.

Submitted by Assigned_Reviewer_3

The authors uses an estimation approach to estimate the scaling matrix and replaces an O(np^+p^3) per iteration cost with an O(np+p^2) cost.

The new optimization method applies to generalized linear models and is extremely well crafted.

The paper is extremely clear, the quality is superb, the originality is good and the significance only limited by the experiments and the importance of generalized linear models.

The experiments are a bit weak.

There are only two examples -- logistic regression and ordinary least squares.

These are compared to general optimization techniques, but not to any methods tailored for these.

There was for example a paper by Byrd, Oztoprak and Nocedal that used sampling estimates of the Hessian for logistic regression.

This is quite closely related to what's done in this paper.

There are also other papers using sampling techniques to estimate the Hessian.

It is a bit disconcerting that the "appendix" is 3 times longer than the paper itself.

In many ways the paper is just an advertising pamphlet for the appendix, but the main body of the paper is well crafted and most readers may not wish to dive into the most horrid derivations in the appendix.

Here are some detailed comments: "Distribution of a random variable belongs to an..." --> "The distribution of a random variable belongs to an..."

I expected to see the dependence of t shown on the RHS in the first equation in chapter 3.

As it stands it's just an unnecessary repetition of 2.3.

"well-approximates": this is not a word.

The "event" E defined in 4.1 is not properly defined.

I imagine this would be a set over \beta^{t} in which the set should be written

\{\beta^{t}: |\mu_2(\beta^{t})+...| > \tau,...\} The way it is written, it could also be a set over tau, beta_*, Sigma etc.

What's the difference between ||\beta^t-\beta_*|| and E[||\beta^t-\beta_*||] in Lemma 4.1, and what is the expectation over? Given that the proof is in the lengthy appendix, an explanation/discussion would be helpful.

Could you define sub-gaussian in the main body of the paper?

Figure 2 is a bit busy and I can't tell which line is which as some of the colors are very close and the legend is not sufficiently good to match the line up with the legend.

The bibliography has not been done in the usual NIPS style.

I think the authors need to conform with the usual standard NIPS template.

Summary: This is a well written paper that introduces the new Newton-Stein method which is a well crafted method for learning parameters in a generalized linear model.

The method is analyzed theoretically with supporting evidence from experiments.

The only draw-back is that 24 pages are located in the appendix.

This really is already a solid journal paper.

Submitted by Assigned_Reviewer_4

The paper proposed a new second order optimization method for Generalized Linear Models (GLMs) with an exponential family. The proposed method efficiently estimates the inverse Hessian/scaling matrix with Gaussian assumption on covariates. Multiple aspects of the method are thoroughly explored such as convergence analysis, accuracy of estimation, parameter tuning. Experimental results are consistent with their analysis and the performance of the method is comparable to popular competitors. The paper is well written and well presented.

I like the overall direction of this paper. Some suggestions/questions are the following.

1) GLMs do not assume that its density is an exponential family. It is more general. Please add related comments to avoid confusion on the scope of this paper.

2) The limited results seem fairly compelling, though I'd like to see more comparisons with competitor algorithms with other GLMs (e.g., Poisson as L284) .

3) Step 2 estimates the \hat{Sigma_S} after subsampling. For the robust estimation, low-rank approximation is a good approach. However, the covariance matrix is fixed and inaccurate estimation will damage the performance of the proposed method.

Discuss the impact of poor estimation of \hat{Sigma_S} and potential solutions.

4) L138 : Unlike high-dimensional case (n << p), in the case of n >> p, O(np) and O(np^2) might not have huge difference. Hessian matrix calculation O(np^2) is already relatively cheap. Why is it the main bottle neck?

Minor issues:

1) L234 Is Bp(R)

define as an open ball centered at the origin? L157 step 2:

does it need no constraint for M to get an invertible matrix zeta?
Summary: The paper proposed an efficient second order optimization method for Generalized Linear Models (GLMs) with an exponential family. It is limited to logistic/linear regressions but it shows good performance and its multiple aspects (convergence analysis, accuracy of estimation, parameter tuning) are thoroughly explored.

Author Feedback
Author rebuttal: We thank the reviewers for their thoughtful comments. Please find our detailed response below.

**Assigned_Reviewer_1:

-Natural Gradient Descent:
There are two different formulations of NGD,
1- First formulation uses the inverse Fisher information matrix as the scaling matrix (also referred as Fisher scoring). In the GLM setting we consider, this reduces to the inverse Hessian which is just Newton's method (already included in the experiments).
2- The second version relies on the sample covariance matrix of the gradients. We will include this version of NGD into our experiments.

-The theory section:
We appreciate the reviewer's constructive suggestions. We agree with the reviewer that moving the sub-gaussian case to the appendix, and instead adding a "proof sketch" would provide more insight to the reader.

**Assigned_Reviewer_3:

1) We borrowed the GLM definition from [McCullagh, Nelder 89] where we consider the canonical links. We will address the reviewer's concern, and clarify this point.

2) We will include additional comparisons with a GLM type that does not satisfy our assumptions (i.e., Poisson).

3) Poor estimation of the covariance matrix \Sigma will result in inaccurate curvature approximation. As the reviewer points out, the performance of the algorithm may degrade. We will discuss this point in detail.

4) We assume that n >> p >> 1, (p >> 1 is missing in line 138!) so that gradient operations are still feasible (O(np) computation), but the Hessian computations are not (O(np^2) computation). Note that when p is also large (along with n>>p), O(np^2) and O(np) will have a huge difference.

Minor issues:
-Bp(R) is a closed ball of radius R centered at the origin. But it can be replaced with any convex, closed set S \subset R^p. Our theorems would remain valid with R replaced with diameter of S divided by 2.

-In the definition of \zeta_r in L157, output of argmin operation is a rank r matrix which is not invertible. After adding \hat{\sigma}^2I, we end up with an invertible matrix. Therefore no additional assumptions are needed to make \zeta_r invertible.

We thank the reviewer for his/her suggestions. We will clarify these points in a revised version of the paper.

**Assigned_Reviewer_4:
-Experiments:
We will include experiments with a GLM type that does not satisfy our assumptions (i.e. Poisson), and a subsampling based method such as Byrd et. al.

-"The dependence of t":
In L143 superscript of \beta is missing. \beta should be replaced with \beta^t.

-"The "event" E": This set is a subset of the underlying probability space. In general, elements of a probability space is denoted by w. In this case, the set would be \{w: |\mu_2(\beta^{t})+...| > \tau,...\}. Note that \beta^t is a random variable, hence it is a real, measurable function of w. We will address the reviewer's concern.

-"The difference between ||\beta^t-\beta_*|| and E[||\beta^t-\beta_*||]":
As the covariates (x_i's) are random vectors, \beta^t is also random where the randomness is introduced through the update equation. The expectation is over the randomness due to covariates.

We thank the reviewer for his/her constructive suggestions. We will clarify these issues in the paper.

**Assigned_Reviewer_7:
In Section 2, we describe the GLM setting for canonical links.
This formulation brings several simplifications where the algorithms like
Newton-Raphson, Fisher scoring and IRLS reduce to the same algorithm. This is discussed in detail in [McCullagh, Nelder 89-p43] and [Friedman, Hastie, Tibshirani 01-p121].

In our experiments, we compare Newton-Stein method with Newton-Raphson which covers all three algorithms mentioned above. We agree that this point needs to be clarified in the paper.